# Targeting colonic macrophages improves glycemic control in high-fat diet-induced obesity

Theresa V. Rohm[1,2], Lena Keller [1,2], Angela J. T. Bosch[1,2], Shefaa AlAsfoor [1,2], Zora Baumann [1,2], Amandine Thomas[3], Sophia J. Wiedemann[1,2], Laura Steiger[1,2], Elise Dalmas[1,2], Josua Wehner[1,2], Leila Rachid[1,2], Catherine Mooser[4,5], Bahtiyar Yilmaz [4,5], Nerea Fernandez Trigo[4,5], Annaise J. Jauch[2], Stephan Wueest [6], Daniel Konrad [6], Sandrine Henri [7], Jan H. Niess[2,8], Petr Hruz[2,8], Stephanie C. Ganal-Vonarburg[4,5], Julien Roux [2,9], Daniel T. Meier [1,2] & Claudia Cavelti-Weder [1,2,10✉]

The obesity epidemic continues to worsen worldwide. However, the mechanisms initiating glucose dysregulation in obesity remain poorly understood. We assessed the role that colonic macrophage subpopulations play in glucose homeostasis in mice fed a high-fat diet (HFD). Concurrent with glucose intolerance, pro-inflammatory/monocyte-derived colonic macrophages increased in mice fed a HFD. A link between macrophage numbers and glycemia was established by pharmacological dose-dependent ablation of macrophages. In particular, colon-specific macrophage depletion by intrarectal clodronate liposomes improved glucose tolerance, insulin sensitivity, and insulin secretion capacity. Colonic macrophage activation upon HFD was characterized by an interferon response and a change in mitochondrial metabolism, which converged in mTOR as a common regulator. Colon-specific mTOR inhibition reduced pro-inflammatory macrophages and ameliorated insulin secretion capacity, similar to colon-specific macrophage depletion, but did not affect insulin sensitivity. Thus, pharmacological targeting of colonic macrophages could become a potential therapy in obesity to improve glycemic control.

[1] Clinic of Endocrinology, Diabetes and Metabolism, University Hospital Basel, Basel, Switzerland. [2] Department of Biomedicine (DBM), University of Basel, University Hospital Basel, Basel, Switzerland. [3] Biozentrum, University of Basel, Basel, Switzerland. [4] Department of Visceral Surgery und Medicine, Bern University Hospital, University of Bern, Bern, Switzerland. [5] Department for BioMedical Research (DBMR), University of Bern, Bern, Switzerland. [6] Division of Pediatric Endocrinology and Diabetology, and Children's Research Center, University Children's Hospital, University of Zurich, Zurich, Switzerland. [7] Centre d'Immunologie de Marseille-Luminy, Aix Marseille Université, INSERM, CNRS, Marseille, France. [8] Clarunis, Department of Visceral Surgery, University Center for Gastrointestinal and Liver Diseases, St. Clara Hospital and University Hospital Basel, Basel, Switzerland. [9] Swiss Institute of Bioinformatics (SIB), Basel, Switzerland. [10] Department of Endocrinology, Diabetology and Clinical Nutrition, University Hospital Zurich (USZ) and University of Zurich (UZH), Zurich, Switzerland. ✉email: claudia.cavelti-weder@usb.ch

Glucose dysregulation and chronic low-grade inflammation are key features of metabolic disease. However, little is known about the initial events that lead to these metabolic disturbances in high-fat diet (HFD)-induced obesity. Since the gut is the primary organ exposed to food antigens and bacteria, the gastrointestinal tract could play a major role in triggering glucose intolerance and inflammation.

So far, altered gut microbiota known as dysbiosis has received particular attention in diet-induced obesity. Dysbiosis is believed to contribute to metabolic dysfunction, inter alia, via bile acid metabolism, production of short-chain fatty acids, and bacterial product leakage[1]. Accordingly, germ-free mice are protected from glucose intolerance and adiposity[2]. Gut immunity could be the missing link between dietary and microbial cues, and glucose homeostasis, as the gut constitutes the body's largest immune system. Changes in adaptive gut immunity have already been described in mice fed a HFD. These changes include reduced Foxp3$^+$ regulatory T cells and increased IFN-γ-producing Th1, CD8$^+$, and IL-17-producing γδ T cells[3]. In terms of innate immunity, TLR4, TNF, and NFkB are known to be up-regulated in the gut upon HFD[4,5]. Additionally, an increased inflammatory tone of colonic macrophages has been reported to precede adipose tissue inflammation in obesity[6]. These observations suggest that gut innate immune cells, in particular macrophages, could play a crucial role in metabolic disease.

Macrophages are the most abundant leukocytes in the healthy gut[7]. However, intestinal macrophages are not a homogenous cell population; rather, they consist of five distinct subpopulations that follow a differentiation trajectory[8,9]. They originate from Ly6C$^{high}$ monocyte precursors, which first differentiate into CCR2$^+$ macrophages. These can be subdivided into pro-inflammatory subpopulations P1 and P2, and into an intermediate stage P3[10]. P3 macrophages are short-lived/non-mature transitional cells that gradually lose their pro-inflammatory phenotype to become CCR2$^-$ anti-inflammatory/resident macrophages, which include subpopulations P4 and P5[8,9]. In the healthy gut, the majority of intestinal macrophages belong to the resident P5 subpopulation, which is characterized by distinct anergy towards typical pro-inflammatory stimuli, such as bacterial products or TLR-ligands, despite avid phagocytic activity[10,11]. During intestinal inflammation such as colitis, this differentiation trajectory is disrupted at subpopulation P2, whereby pro-inflammatory macrophages accumulate in the gut[9,10,12]. However, it is currently unknown whether obesity affects intestinal macrophage subpopulations, and how they regulate glucose homeostasis. Therefore, we aimed to elucidate the role of colonic macrophage subpopulations in glucose metabolism. Understanding this relationship in-depth might open up innovative immune-modulatory treatments for metabolic disease.

## Results

### Pro-inflammatory colonic macrophage subpopulations increase with high-fat diet.

There is growing evidence that the intestinal immune system is an important contributor to metabolic disease[13]. However, the role played by specific intestinal macrophage subpopulations in glucose homeostasis remains unknown. To address this, we analyzed five distinct colonic macrophage subpopulations (P1-P5) in response to coconut-based HFD, in comparison to mice in a control group fed a chow diet (Fig. 1a). When mice were kept on HFD for up to 12 weeks, mice developed glucose intolerance and hyperinsulinemia already within 1 week, while their body weight was initially unchanged (Fig. 1b and Supplementary Fig. 1a). Concurrent with these metabolic changes, we found increased gene expression of inflammatory/monocyte markers Tnf and Ly6c in colon tissue, while typical markers of resident

macrophages such as Tgfb1, Adgre1, and Cd68 had decreased (Fig. 1c). Subsequently, we investigated whether this early response was reflected by changes in colonic macrophage subpopulations upon HFD. Indeed, we found that 1 week of HFD increased CCR2$^+$ pro-inflammatory colonic macrophages, especially in the P1 and P2 subpopulations, which persisted for up to 12 weeks of HFD, compared to the chow diet control group (Fig. 1d, Supplementary Fig. 1b, c). In contrast, in CCR2$^-$ anti-inflammatory/resident macrophages, mainly the subpopulation P4 was reduced. In addition, colonic neutrophils increased at all time points, while eosinophils and dendritic cells did not show a consistent pattern over the time course of three months (Supplementary Fig. 2a–c). Stomach size was reduced after 4 and 12 weeks of HFD, while the cecum and colon were already shortened after 1 week of HFD (Supplementary Fig. 2d), a finding which is consistent with the previous research[6]. Adipose tissue inflammation in terms of macrophage accumulation, a hallmark of chronic inflammation in metabolic disease, appeared later at 12 weeks of HFD as shown by increased pro-inflammatory gene expression and adipose tissue macrophage (ATM) subpopulation M1b (Fig. 1e, f and Supplementary Fig. 1d). Concomitant with altered glucose metabolism and gut innate immunity, plasma TNF and IL-6 were increased from 1 week of HFD feeding onwards (Fig. 1g). These data indicated that pro-inflammatory colonic macrophage subpopulations increased with HFD, a response that occurred simultaneously with glucose intolerance but before obesity and adipose tissue inflammation.

### Gut microbiota are essential for increases in pro-inflammatory colonic macrophage subpopulations upon HFD, while the fat source modulates their magnitude.

We next investigated whether the fiber content or fat source influence colonic macrophage subpopulations and glucose metabolism. Mice given fiber-rich chow were compared to those given fiber-deficient starch control diet and mice fed coconut- and lard-based HFDs to their respective controls (coconut-HFD vs. chow; lard-HFD vs. starch control diet) (Supplementary Table 1). Such a detailed assessment is crucial as regular chow contains fiber and may differ considerably depending on seasonal harvests, unlike purified/standardized HFDs. Body weights increased similarly in the two control groups and in the HFD groups (Supplementary Fig. 3a). Both control diets, independent of fibers being present, did not impair glucose metabolism, while mice fed HFD developed glucose intolerance, hyperinsulinemia, and insulin resistance (Fig. 2a). This was seen particularly with those on a lard-based HFD. Comparing mice on the chow and starch diet, we found that the fiber content did not affect colonic macrophage or dendritic cell subpopulations, fat pad weights, M1 ATMs, or plasma levels of TNF and IL-6 (Supplementary Fig. 3b–f). When comparing mice on the two different HFDs to mice on their respective control diets (coconut-HFD vs. chow; lard-HFD vs. starch control diet), both HFD groups had an increase in pro-inflammatory CCR2$^+$ colonic macrophages, especially in the P1 and P2 subpopulations (Fig. 2b and Supplementary Fig. 3b). Additionally, mice on lard-based HFD had an increased total number of colonic macrophages (Fig. 2b and Supplementary Fig. 3b). Colonic dendritic cells did not show a consistent pattern upon HFD (Supplementary Fig. 3d, e). While mice on lard-based HFD showed higher total ATMs, both HFD groups had increased M1b ATMs, plasma TNF, and IL-6 (Fig. 2c, Supplementary Fig. 3c, f). Thus, fiber content did not impact colonic macrophages, while the source of dietary fat modulated the magnitude of the pro-inflammatory innate immune response and glycemic control.

Next, we used 16S ribosomal gene sequencing to assess how the different diets (coconut-HFD vs. chow; lard-HFD vs. starch control diet) affect gut microbiota as a potential mediator of glucose intolerance. At baseline, microbiota composition was similar

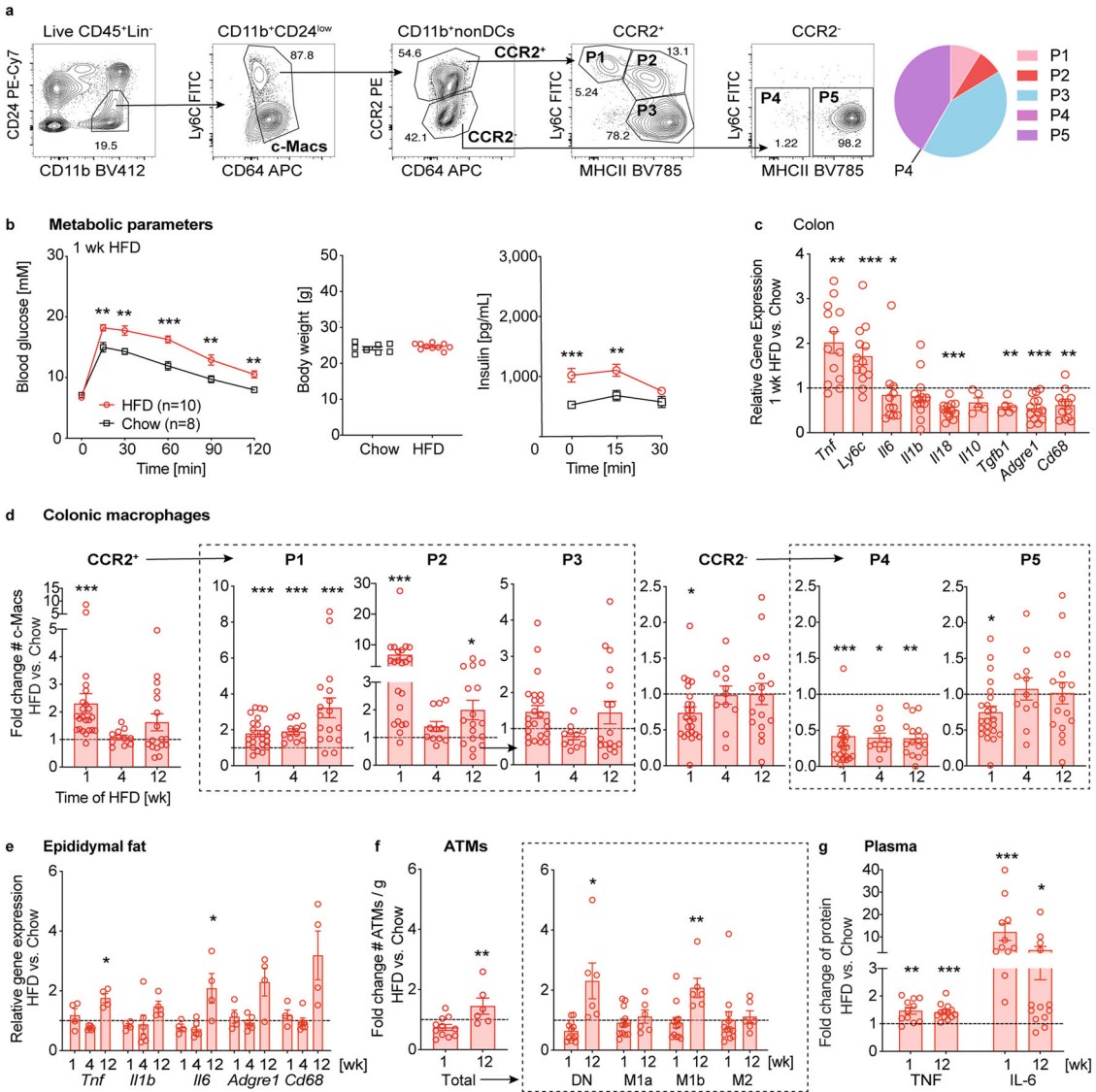

**Fig. 1 Pro-inflammatory colonic macrophage subpopulations increase with high-fat diet.** Wild-type mice were fed either a coconut-based HFD (red circles) or a control diet (black square) for up to 12 weeks: **a** Representative flow cytometry plots and distribution of colonic macrophage (c-Mac) subpopulations P1–P5 in a mouse fed chow. **b** Intraperitoneal glucose tolerance test (IPGTT), body weight, and insulin after 1 week (wk) HFD ($n = 10$) or chow ($n = 8$). **c** Gene expression in colon tissue of mice fed a HFD for 1 wk relative to controls. **d** Fold change in absolute numbers (♯) of monocyte-derived CCR2$^+$ (pro-inflammatory P1, P2, intermediate P3) and anti-inflammatory/resident CCR2$^-$ (P4-P5) c-Macs. **e** Gene expression in adipose tissue of HFD-fed mice relative to controls. **f** Fold change of adipose tissue macrophages (#ATMs/g) and their subpopulations (double negative DN, M1a, M1b, M2) upon HFD. **g** Fold change of plasma TNF and IL-6 upon HFD. Statistical data are expressed as mean ± SEM. Data are representative of one experiment (**b**) or 2–6 (**c–g**) independent experiments, with each data point representing an individual mouse. *$p < 0.05$, **$p < 0.01$, ***$p < 0.001$, unpaired Mann–Whitney $U$ test with two-tailed distribution.

in all groups. Feeding mice fiber-deficient diets (starch control, coconut- and lard-based HFDs) for one week resulted in similar microbial shifts as shown by the Principal Coordinate Analysis and relative abundance at phylum and genus level (Fig. 2d and Supplementary Fig. 3g). After 13 weeks on the diet, however, distinct differences between the two HFDs became apparent: Whereas mice fed a coconut-based HFD had a microbiota composition similar to that at 1 week (reduced *Firmicutes*), those fed a lard-based HFD shifted in the Principal Coordinate Analysis due to changes at the phylum (increase in *Firmicutes* and reduction in *Bacteroidetes*) and genus levels (Fig. 2d and Supplementary Fig. 3g). Hence, microbial dysbiosis per se, but not a specific microbial composition, correlated with glucose intolerance.

To test how HFD affects intestinal macrophage subpopulations and glucose tolerance in the absence of gut microbiota, we used

germ-free mice, which were reportedly protected from glucose intolerance and adiposity[2]. We found partial protection from glucose intolerance with less pronounced hyperinsulinemia in germ-free mice fed a lard-based HFD, compared to the corresponding starch diet control group (Fig. 2e). As previously described, germ-free mice on control diet had fewer colonic macrophages than did colonized mice[14], especially when on a fiber-deficient starch control diet (Supplementary Fig. 4a, c). However, the colonic macrophage subpopulation distribution in germ-free mice was similar to that in colonized mice on a chow diet (Supplementary Fig. 4b). Upon HFD in germ-free mice, there was no increase in pro-inflammatory colonic macrophage P1 and P2 subpopulations, M1b ATMs (except in frequency), plasma TNF, and IL-6, nor changes in dendritic cells (Fig. 2f, Supplementary Fig. 4c–e and Supplementary Fig. 4g, h). Of note, germ-free mice

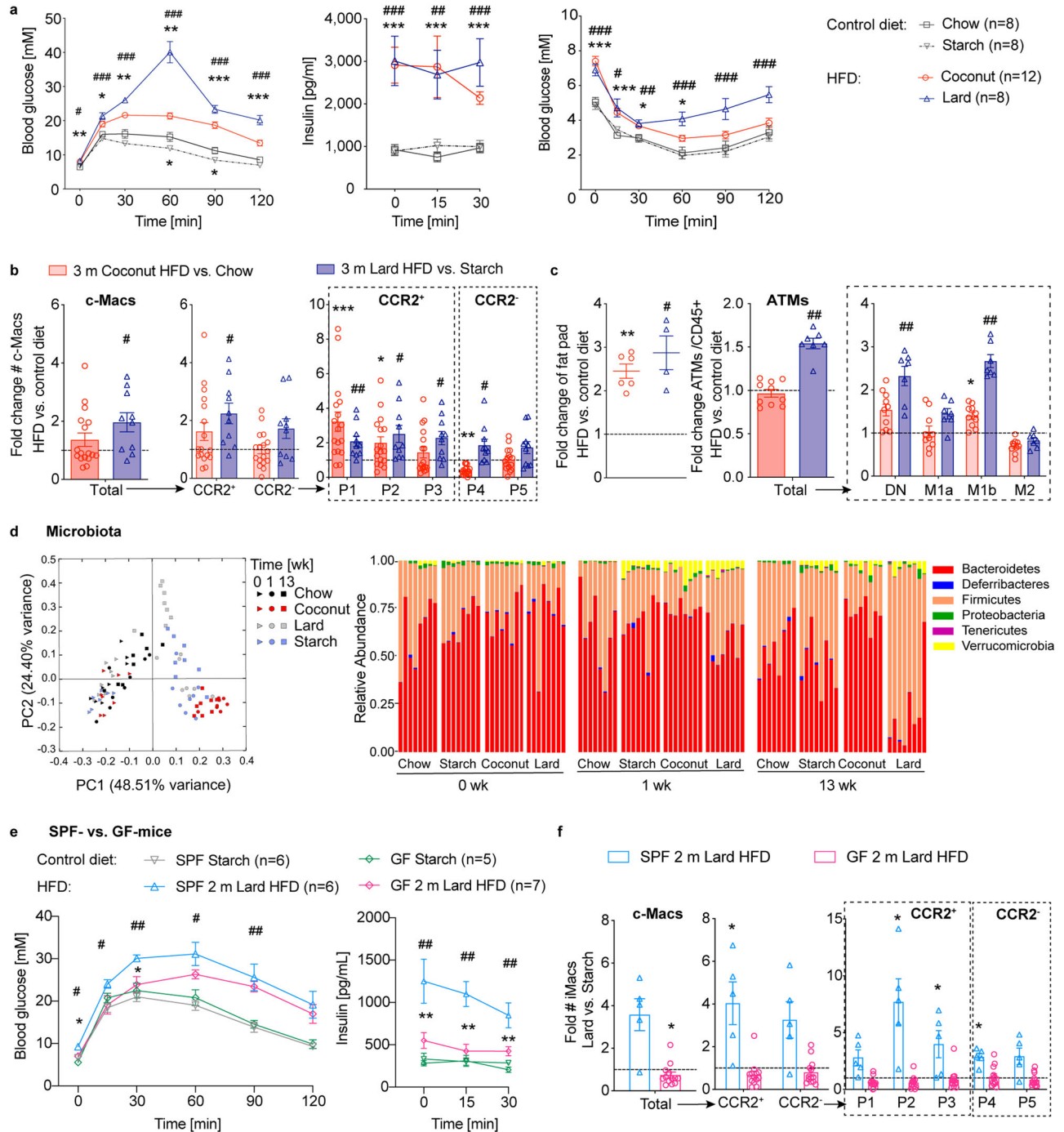

**Fig. 2 Gut microbiota are essential for increases in pro-inflammatory colonic macrophages upon HFD, while the fat source modulates their magnitude.**
For 3 months (m), wild-type mice were fed either a coconut-based HFD (red circles), a lard-based HFD (blue triangles) or a control diet with fibers (chow, black squares) or without fibers (starch, gray triangles). **a** Intraperitoneal glucose tolerance test (IPGTT), insulin and insulin tolerance test (ITT) (starch $n = 8$, chow $n = 8$, coconut $n = 12$, lard $n = 8$). **b**, **c** Fold change of colonic macrophages (c-Macs) (**b**), fat pad weights, and adipose tissue macrophages (ATMs) (**c**), compared to respective controls (coconut-HFD vs. chow; lard-HFD vs. starch). **d** Principal coordinate analysis (PCoA; left) and relative phyla abundances (right) of fecal microbiota before and after 1 week (wk) and 3 m of HFD compared to controls ($n = 8$ per group). Specific pathogen-free (SPF) or germ-free (GF) mice were fed for 2 m either a lard-based HFD (SPF: blue triangles, GF: pink rhombus) or a starch diet (SPF: gray triangles, GF: green rhombus): **e**, **f** IPGTT, insulin (HFD: SPF $n = 6$, GF $n = 7$, starch: SPF $n = 6$, GF $n = 5$) (**e**) and fold change of c-Macs compared to starch controls (**f**). Statistical data are expressed as mean ± SEM. Data are representative of one experiment (**a**, **c**, **d**, **e**, **f** GF in parallel to SPF), two (**b**), or three independent experiments (**f** GF), with each data point representing an individual mouse. **a**, **b**: Coconut-HFD vs. chow: *$p < 0.05$, **$p < 0.01$, ***$p < 0.001$. Lard-HFD vs. starch: #$p < 0.05$, ##$p < 0.01$, ###$p < 0.001$, **e** SPF mice fed lard-HFD vs. GF mice fed lard-HFD: *$p < 0.05$, **$p < 0.01$. SPF mice fed lard-HFD vs. starch: #$p < 0.05$, ##$p < 0.01$, unpaired Mann–Whitney $U$ test with two-tailed distribution.

showed very low numbers of CD11b$^+$CD103$^+$ conventional DCs, especially for mice on a lard-based HFD (Supplementary Fig. 4e–g), possibly due to lower TLR activation in the absence of microbial entry[15]. These data indicated that the lack of gut microbiota in germ-free mice prevented them from recruiting and hence increasing pro-inflammatory colonic macrophages upon HFD. This potentially protected them from glucose intolerance.

**Macrophage numbers are linked to glycemic control.** As the degree of innate inflammatory response correlated with glycemic control, we next hypothesized that the number of macrophages is directly linked to glucose metabolism. To assess whether a dose-dependent macrophage depletion would gradually improve glycemic control, we depleted macrophages in a dose-dependent manner by treating mice fed a coconut-based HFD with increasing doses of the colony-stimulating factor 1 receptor (CSF1R)-inhibitor BLZ945 (50, 100, 200 µg/g/d) or its vehicle. CSF1R regulates the survival, pro-liferation, and differentiation of macrophages[16]. Indeed, glucose metabolism gradually improved by increasing BLZ945 concentrations, as shown by fasting glycemia, glucose tolerance, and insulin sensitivity (Fig. 3a and Supplementary Fig. 5a). Body weights were slightly reduced, with the highest dose after 2 months of CSF1R-inhibition, potentially reflecting lower insulin levels (Supplementary Fig. 5a, b). Concerning gut innate immunity, treatment with BLZ945 reduced CCR2$^+$ pro-inflammatory and CCR2$^-$ anti-inflammatory/resident colonic macrophages and their P1-P5 subpopulations, with the most pronounced effects seen in the mid and high dose groups (Fig. 3b and Supplementary Fig. 5c). Additionally, CSF1R-inhibition resulted in a dose-dependent reduction in large and small peritoneal macrophages, microglia, and ATMs (after 2 weeks preferentially M2 ATMs and after 2 months additionally M1 ATMs) (Fig. 3c–e and Supplementary Fig. 5d). Thus, these results suggested a direct link between the number of macrophages and glycemic control. However, as our dose-titration with increasing doses of a CSF1R-inhibitor affected multiple tissue macrophages, a colon-specific macrophage depletion model is needed to assess the causal link between colonic macrophages and glycemic control.

**Depleting colonic macrophages improves glucose metabolism.** We devised a pharmacological approach to colon-specifically deplete macrophages. To this end, we administered clodronate liposomes intrarectally and assessed whether colon-specific macrophage deple-tion is feasible and how it affects glucose homeostasis. Indeed, intrarectal clodronate liposomes in mice fed a coconut-based HFD resulted in improved fasting glycemia and glucose tolerance without affecting body weight (Fig. 4a, left panel). Despite having lower glucose levels, mice treated with intrarectal clodronate showed a trend towards enhanced insulin secretion, suggesting improved β-cell function (Fig. 4a, middle panel). Additionally, mice treated with intrarectal clodronate had improved insulin sensitivity, as seen in the insulin tolerance test (Fig. 4a, right panel). Ex vivo glucose-stimulated insulin secretion of isolated islets from clodronate-treated mice indicated improved β-cell function, as shown by lower basal insulin and an enhanced stimulation index (Fig. 4b). These metabolic changes were associated with a reduction in colonic macrophages in the proximal colon, especially pro-inflammatory P1 and P2 subpopulations (Fig. 4c and Supplementary Fig. 6a). Meanwhile, other tissue macrophages were not reduced, such as in adipose tissue, liver, pancreatic islets, and brain tissue (Fig. 4d, e and Supplementary Fig. 6b, c). Pro-inflammatory cytokines were not increased in colon tissue or plasma of mice intrarectally treated with clodronate lipo-somes, excluding confounding effects of local or systemic inflam-mation, as have been observed with systemic clodronate liposome administrations[17] (Supplementary Fig. 6d, e). Also, microbial dys-biosis was ruled out as a possible mediator for improved glycemic

control since microbiota composition was not altered in mice treated with clodronate (Fig. 4f and Supplementary Fig. 6f).

Clodronate liposomes given intraperitoneally served as a control for systemic macrophage depletion. Thereby, intraperitoneal administrations of clodronate liposomes reduced macrophages in multiple tissues such as the colon, adipose tissue, and liver, while islet macrophages and microglia were unchanged (Supplementary Fig. 7a–c). Plasma IL-6 was elevated by systemic clodronate liposomes as previously reported (Supplementary Fig. 7d)[17]. Systemic macrophage depletion by clodronate liposomes improved glucose tolerance and insulin sensitivity as indicated by lower circulating insulin levels in mice on HFD (Supplementary Fig. 7e). Overall, colon-specific macrophage depletion in mice improved their glucose tolerance due to enhanced insulin sensitivity and β-cell function. Thus, these findings established a causal link between the number of colonic macrophages and glycemic control.

**Colonic macrophages upon HFD show an interferon signature and altered mitochondrial metabolism with activated mTOR as a common regulator.** Next, we addressed the question of what factors might mediate the cross-talk between intestinal macrophages and β-cell function. First, we assessed the role GLP-1 played as enteroendocrine cells have been suggested to orchestrate mucosal immunity. Inhibiting GLP-1 activity with exendin (9–39) before glucose stimulation in vivo did not fully revert improved glycemic control in macrophage-depleted mice (Supplementary Fig. 7f), suggesting that improved β-cell function might be partially mediated through GLP-1. Although, systemic levels of GLP-1 were not enhanced in HFD mice after colonic macrophages were depleted by intrarectal clodronate liposomes (Supplementary Fig. 7f). Next, we tested whether the parasympathetic nervous system could link gut innate immunity and β-cell function. However, blocking the para-sympathetic activity by atropine prior to glucose tolerance testing in macrophage-depleted mice also did not fully abolish improved glycemic control (Supplementary Fig. 7g).

To narrow down the focus on signaling pathways in intestinal macrophages, which might give reference to the cross-talk between colonic macrophages and β-cells, we performed single-cell RNA-sequencing (scRNA-seq) of colonic macrophages upon coconut-based HFD feeding. Hierarchical clustering analysis identified the macrophage P1–P5 subpopulations as previously defined by flow cytometry, whereby the P2 and P5 subpopulations comprised two related clusters (Fig. 5a–c). This classification was consistent with ab initio annotation by using ImmGen FACS-sorted bulk samples of pure cell types as a reference[18] (Supplementary Fig. 8a). Cell clusters positioned along a presumed differentiation trajectory (PC1), and an activation/inflammation axis (PC2), showed little overlap between chow diet and HFD, which indicated a strong transcriptional response upon HFD (Supplementary Fig. 8b). Consistent with flow cytometry, upon HFD there was a relative increase in pro-inflammatory subpopulations and a decrease in anti-inflamma-tory/resident subpopulations (Fig. 5b). To compare the transcrip-tomes of colonic macrophages between chow diet and HFD, we stratified the analysis into subpopulations (clusters) to correct for differential abundance across conditions. Based on the number of differentially expressed genes, HFD induced the most pronounced effects in colonic macrophage P1 and P2 subpopulations (Supple-mentary Fig. 8c). Genes up-regulated by HFD involved interferon signaling (*Irf7, Ifitm1, Ifi205, Isg15*), chemokines (*Cxcl9, Ccl5/8*), guanylate-binding proteins (*Gbp2/5*), and macrophage activation genes (*Ly6a/c2/i*) (Supplementary Fig. 8c and Supplementary Fig. 9).

In addition, analysis of sets of regulatory transcription factors, termed regulons revealed enhanced *Stat1/2, Irf1/8,* and *Etv7* under HFD, the latter as a component of the mTOR complex mTORC3[19] (Supplementary Fig. 8d). Performing gene set

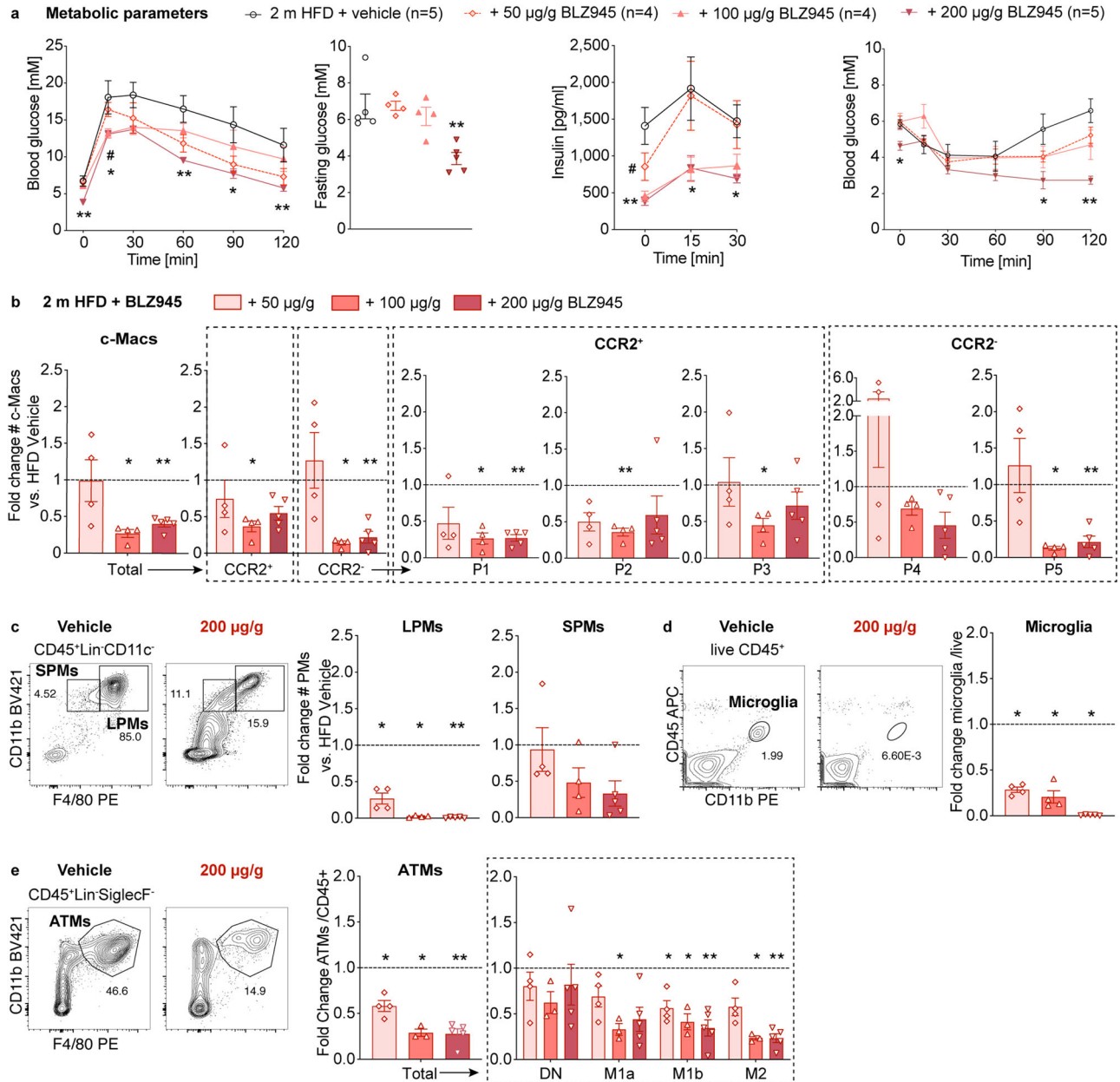

**Fig. 3 Macrophage numbers are linked to glycemic control.** Wild-type mice were fed a coconut-based HFD for 2 months (m) and treated with the CSF1R inhibitor BLZ945 or its vehicle (vehicle n = 5: black circles, 50 μg/g n = 4: light red rhombus, 100 μg/g n = 4: light red triangles, 200 μg/g n = 5: dark red triangles): **a** Intraperitoneal glucose tolerance test (IPGTT), fasting blood glucose, insulin and insulin tolerance test (ITT). **b** Fold change of colonic macrophages (c-Macs). **c-e** Representative flow cytometry plots and fold change of peritoneal macrophages (small: SPM/ large: LPM) (**c**), microglia (**d**), and adipose tissue macrophages (ATMs) (**e**). Statistical data are expressed as mean ± SEM. Data are representative of one (**a-e**), with each data point representing an individual mouse. *p < 0.05, **p < 0.01, ***p < 0.001, unpaired Mann–Whitney U test with two-tailed distribution.

enrichment analysis by using MSigDB hallmark pathways showed up-regulation of interferon-gamma (IFN-γ) and alpha (IFN-α) response, oxidative phosphorylation, and allograft rejection in mice fed a HFD (Fig. 5d and Supplementary Fig. 8e). We confirmed an altered mitochondrial metabolism in these macrophages by increased measures of mitochondrial mass, membrane potential, and abundance of reactive oxygen species in all colonic macrophage subpopulations of mice fed a HFD (Fig. 5e). As mTOR is a common regulator of both interferon signaling and energy metabolism[20,21], we postulated that altered mTOR signaling might be involved in the transcriptional changes in colonic macrophages upon HFD. Indeed, in all macrophage subpopulations of mice fed a HFD, we found mTOR activity to be

enhanced, as shown by increased phosphorylation of S6 (pS6) and Akt (pS473) (Fig. 5f). Hence, enhanced mTOR signaling could mediate the transcriptional response in macrophages under HFD, involving an interferon signature and a change in mitochondrial metabolism.

**Colon-specific mTOR inhibition improves insulin secretion capacity.** Next, we assessed whether enhanced mTOR signaling in the gut contributes to HFD-related glucose intolerance. Therefore, we intrarectally administered the mTOR inhibitor rapamycin to mice on a coconut-based HFD and assessed the effect it had on glucose metabolism in vivo. We confirmed that pS6 and pS473 were

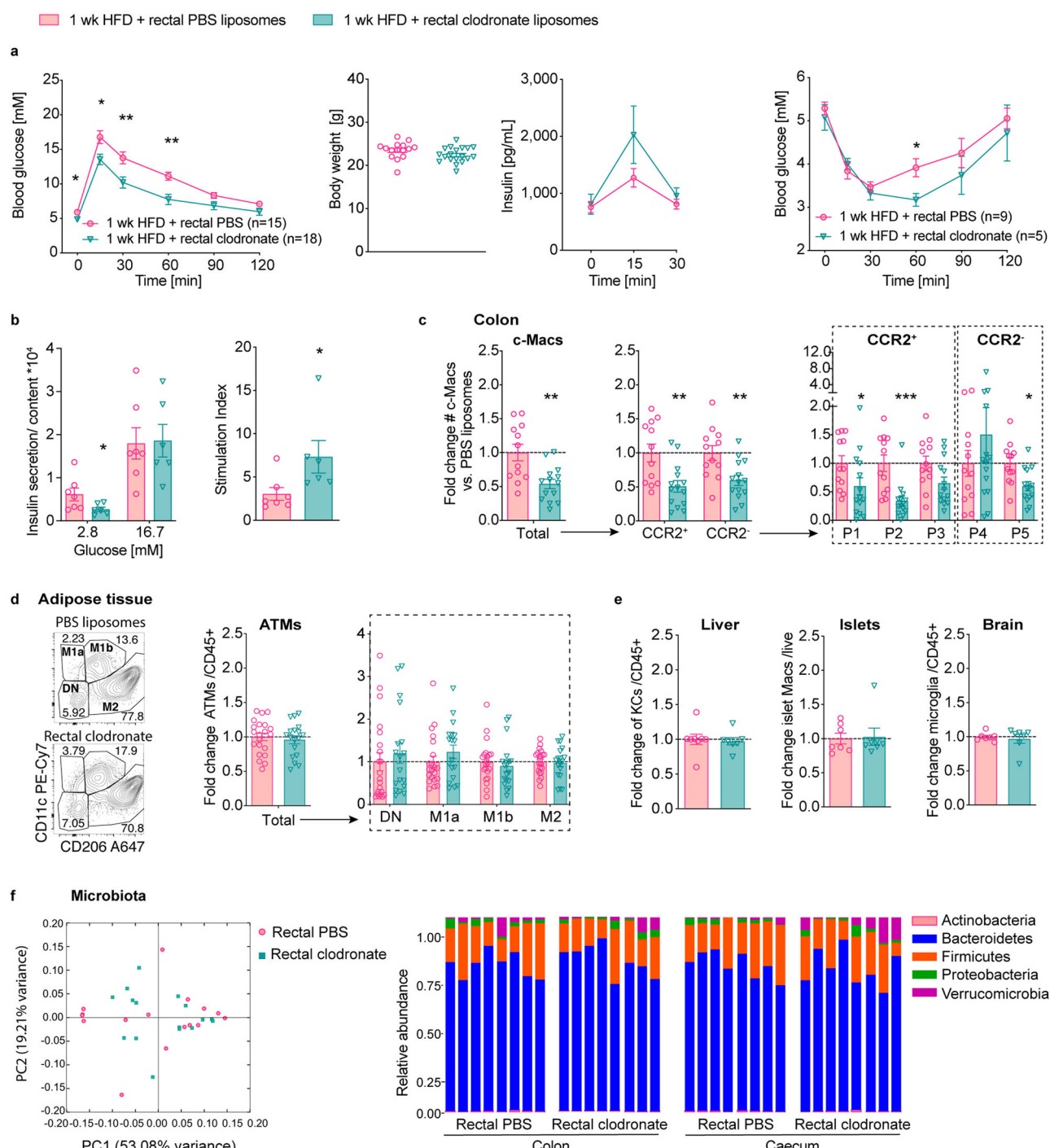

**Fig. 4 Depleting colonic macrophages improves glucose metabolism.** Wild-type mice were fed a coconut-based HFD for 1 week (wk) and treated intrarectally with clodronate (turquoise triangles) or PBS liposomes (pink circles): **a** Intraperitoneal glucose tolerance test (IPGTT), body weight at IPGTT (PBS n = 15, clodronate n = 18), insulin (PBS n = 9, clodronate n = 11), and insulin tolerance test (ITT) (PBS n = 9, clodronate n = 5). **b** Basal and glucose-stimulated insulin secretion in islets ex vivo. **c** Fold change of colonic macrophages (c-Macs) in the proximal colon. **d** Representative flow cytometry plots and fold change of adipose tissue macrophages (ATMs). **e** Fold change of Kupffer cells (KCs), islet macrophages, and microglia. **f** Principal coordinate analysis (PCoA; left) and relative phyla abundances (right) of fecal microbiota (n = 8 per group). Statistical data are expressed as mean ± SEM. Data are representative of one (**a** ITT, **b**) experiment or five (**a** GTT), three (**c**, **d**), two (**e**, **f**) independent experiments, with each data point representing an individual mouse. *p < 0.05, **p < 0.01, ***p < 0.001, unpaired Mann–Whitney U test with two-tailed distribution.

decreased in all colonic macrophage subpopulations upon intrarectal rapamycin treatment (Fig. 6a). In addition, colon-specific mTOR inhibition reduced the superoxide (ROS) indicator MitoSOX in all colonic macrophages, which suggests decreased mitochondrial metabolism (Fig. 6b). Tissue macrophages derived from adipose tissue, spleen, or the peritoneal cavity did not show a reduction in mTOR activity (Fig. 6c–e). Intrarectal rapamycin did not affect body weight, whole-body glucose tolerance, or insulin sensitivity but increased stimulated insulin levels (Fig. 6f). Islets from mice intrarectally treated with rapamycin showed lower basal insulin and an

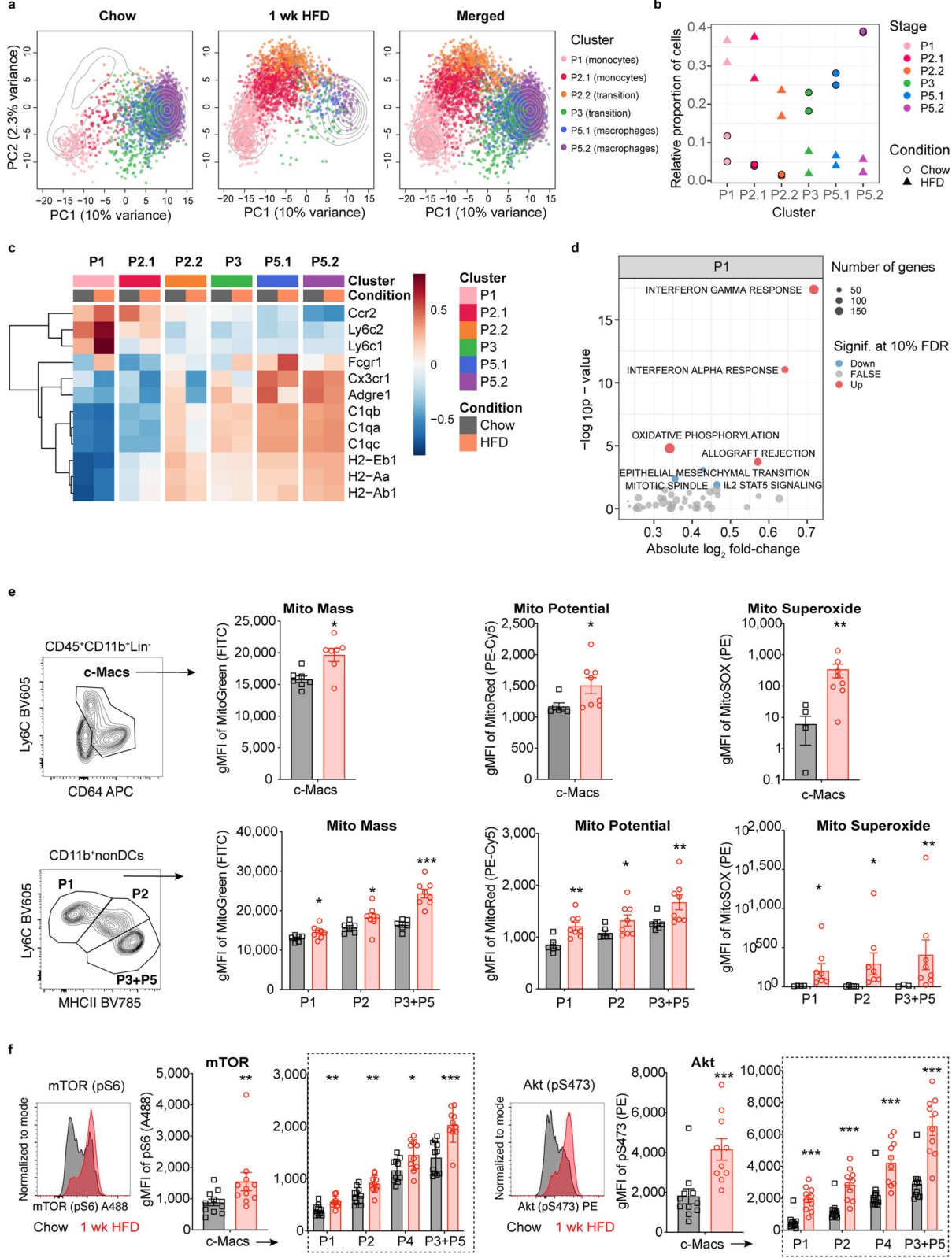

increased stimulation index ex vivo as a measure for improved β-cell function (Fig. 6g), similar to mice treated with intrarectal clodronate liposomes.

In contrast, systemic rapamycin by intraperitoneal injections led to pronounced glucose intolerance, and insulin resistance in mice fed a HFD, as previously reported[22] (Supplementary Fig. 10a). As a measure of mTOR activity, pS6 and pS473 were not only reduced in colonic macrophages of mice intraperitoneally treated with rapamycin but also in ATMs and splenic monocytes (Supplementary Fig. 10b–d). Further, systemic mTOR inhibition was associated with reduced M2 ATMs (Supplementary Fig. 10e). These findings demonstrated that colon-specific inhibition of mTOR improved

**Fig. 5 Colonic macrophages upon HFD show an interferon signature and altered mitochondrial metabolism with activated mTOR as a common regulator.** Wild-type mice were fed a coconut-based HFD for 1 week (wk) and compared to mice fed chow: **a, b** Principal component analysis (PCA) (**a**) and relative proportion of colonic macrophages (c-Macs) (**b**). **c** Average gene expression of specific marker genes in c-Mac clusters. **d** Up- (red) or down-regulated (blue) Molecular Signatures Database (MSigDB) hallmark pathways after 1 wk of HFD for cluster P1 (FDR < = 0.1). **e** Geometric mean fluorescent intensity (gMFI) of mitochondrial mass (MitoGreen), potential (MitoRed), and reactive oxygen status (MitoSOX) of total c-Macs and their subpopulations from mice fed 1 wk a HFD (red circles) compared chow-fed controls (black squares). **f** mTOR activation was measured by gMFI of phosphorylated S6 (pS6) and Akt (pS475) in c-Macs. Statistical data are expressed as mean ± SEM. Data are representative of two replicates ($n = 2$ per group) (**a–d**) and one (**e**) or two (**f**) independent experiments, with each data point representing one cell (a) or one individual mouse (**e–g**). *$p < 0.05$, **$p < 0.01$, ***$p < 0.001$, unpaired Mann–Whitney $U$ test with two-tailed distribution.

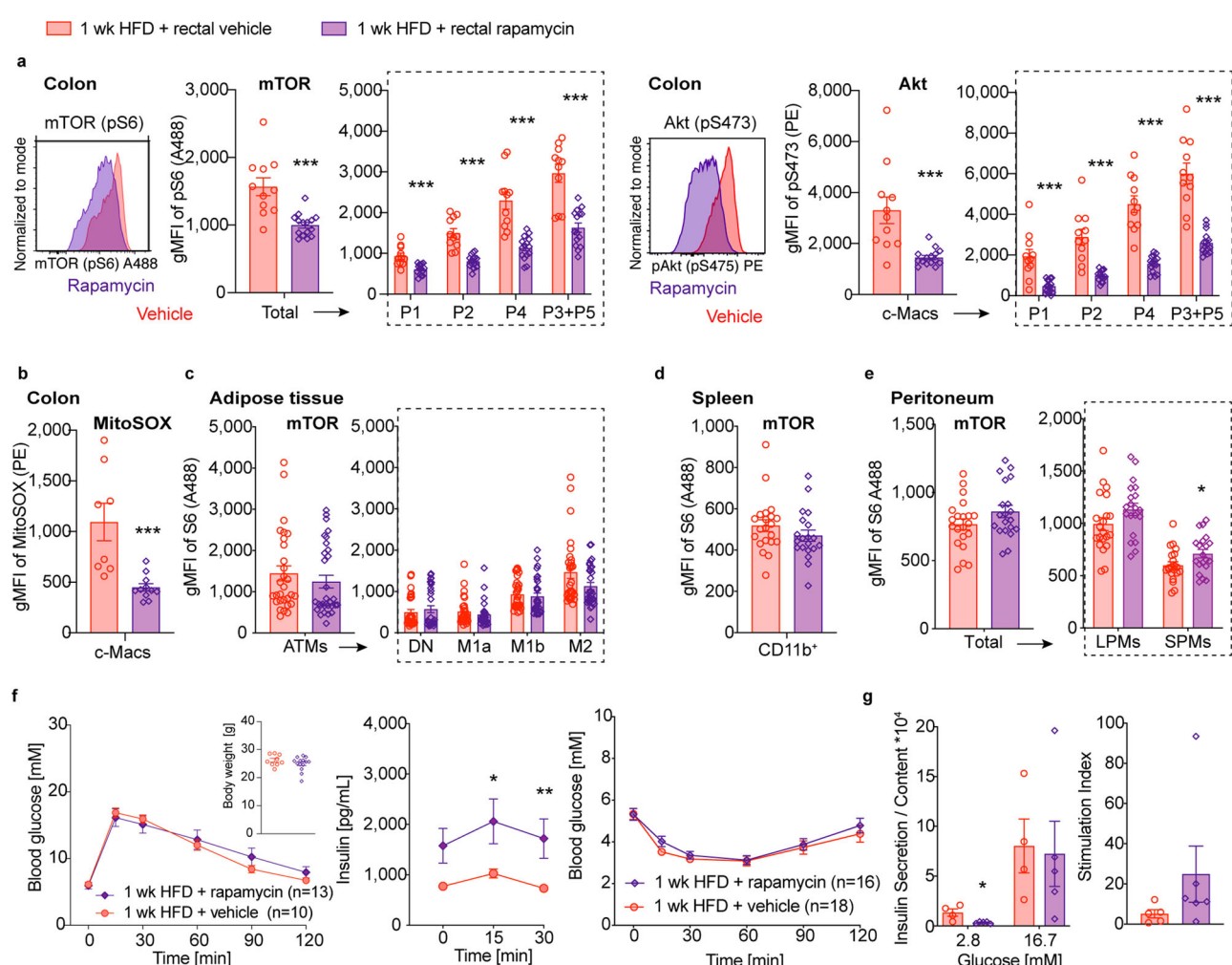

**Fig. 6 Colon-specific mTOR inhibition improves insulin secretion capacity.** Wild-type mice were fed a coconut-based HFD for 1 week (wk) and treated intrarectally with 3 mg/kg rapamycin (purple rhombus) or vehicle (red circles): **a** Geometric mean fluorescent intensity (gMFI) of phosphorylated S6 (pS6) and Akt (pS473) in colonic macrophages (c-Macs). **b** gMFI of MitoSOX in c-Macs. **c–e** gMFI of pS6 in adipose tissue macrophages (ATMs) (**c**), splenic monocytes (**d**) and peritoneal macrophages (small: SPM/large: LPM) (**e**). **f** Intraperitoneal glucose tolerance test (IPGTT: vehicle $n = 10$, rapamycin $n = 13$), body weight, and insulin levels at IPGTT, and insulin tolerance test (ITT: vehicle $n = 18$, rapamycin $n = 16$). **g** Basal and glucose-stimulated insulin secretion (GSIS) in islets ex vivo. Data are representative of one (**b, g**) experiment or two (**a, f** GTT), three (**d–f** ITT), or four (**c**) independent experiments, with each data point representing one individual mouse. *$p < 0.05$, **$p < 0.01$, ***$p < 0.001$, unpaired Mann–Whitney $U$ test with two-tailed distribution.

insulin secretion capacity, while systemic mTOR inhibition induced pronounced insulin resistance.

**While elevated pro-inflammatory colonic macrophages are linked to impaired β-cell function, colonic mTOR inhibition restores colonic macrophages and insulin secretion capacity.** To examine the functional activity of colonic macrophages upon HFD, and after additional mTOR inhibition, we characterized the

cytokine profile (TNF, IL-1β, IL-6, IL-10) in macrophages from mice fed a coconut-based HFD or a chow diet for 1 week. We found more TNF+, IL-1β+, IL-6+ and IL-10+ colonic macrophages in mice fed a HFD, especially in the pro-inflammatory P2 subpopulation after LPS stimulation (Fig. 7a). In contrast, when mice on a HFD were additionally treated with rapamycin by intrarectal administrations, colonic mTOR inhibition led to reduced cytokine expression (gMFI) of TNF and IL-1β in pro-inflammatory colonic macrophages (Fig. 7b). Additionally, rectal

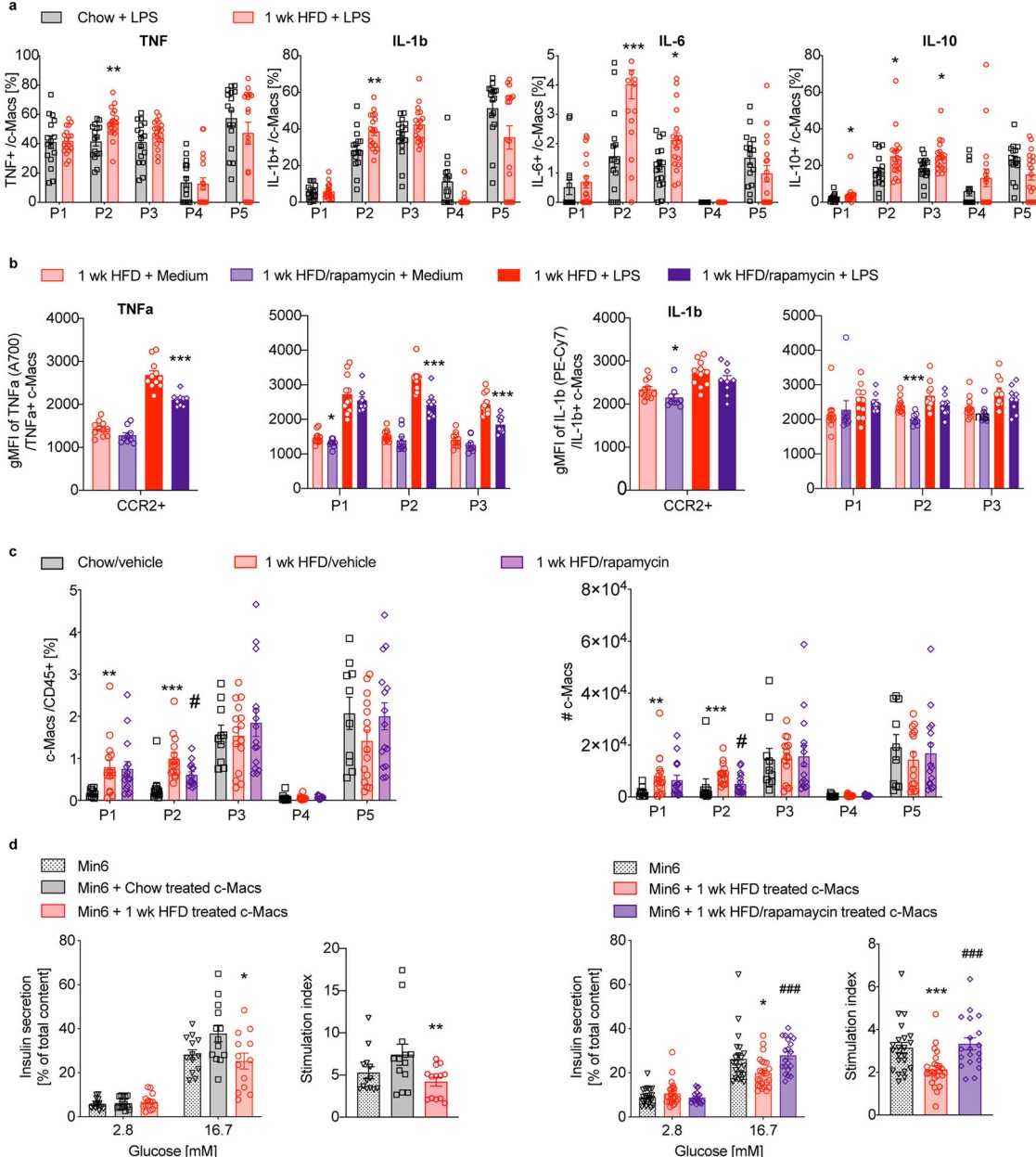

**Fig. 7 While elevated pro-inflammatory colonic macrophages are linked to impaired β-cell function, colonic mTOR inhibition restores colonic macrophages and insulin secretion capacity.** Wild-type mice fed a coconut-based HFD (red circles) for 1 week (wk) were compared to chow controls (black squares) or HFD-fed mice treated with 3 mg/kg intrarectal rapamycin compared to vehicle controls (purple rhombus): **a** Frequency of cytokine-producing colonic macrophages (tumor necrosis factor (TNF)⁺, interleukin-1b (IL-1b)⁺, IL-6⁺ and IL-10⁺ c-Macs) after 4.5 h culture with 1 μg/mL lipopolysaccharide (LPS). **b** Geometric mean fluorescent intensity (gMFI) of TNF and IL-1b of cytokine-producing c-Macs. **c** Frequency and absolute numbers (♯) of c-Macs isolated from chow or HFD mice treated with 3 mg/kg rapamycin or vehicle. **d** Basal and glucose-stimulated insulin secretion (GSIS) in Min6 cells after 16 h co-culture with sorted c-Macs. Data are representative of two (**a**, **b**, **d** left panel) or three (**c**, **d** right panel) independent experiments, with each data point representing one individual mouse (**a**–**c**) or one well (**d**). *$p < 0.05$, **$p < 0.01$, ***$p < 0.001$, unpaired Mann–Whitney $U$ test with two-tailed distribution.

mTOR inhibition in HFD-fed mice also affected the number of macrophages as the pro-inflammatory macrophage subpopulation P2 was reduced in the colon (Fig. 7c).

To investigate the potential contribution that these cytokines might have on insulin secretion, we co-cultured the β-cell line Min6 with sorted primary colonic macrophages. In Min6 cells co-cultured with colonic macrophages isolated from mice fed a HFD, we found reduced glucose-stimulated insulin secretion and insulin stimulation index (Fig. 7d, left panel). In contrast,

co-culture of Min6 cells with colonic macrophages isolated from HFD-fed mice intrarectally treated with rapamycin (compared to controls treated with vehicle) showed restored glucose-stimulated insulin secretion (Fig. 7d, right panel). This indicated that colonic macrophages from mice fed a HFD had a deleterious effect on β-cell function, while mTOR inhibition by intrarectal rapamycin administration dampened both the number and cytokine profile of pro-inflammatory colonic macrophages, and restored glucose-stimulated insulin secretion.

## Discussion

Our results revealed direct cross-talk between nutritional cues, colonic macrophage subpopulations, and glucose homeostasis. Importantly, our study distinguished specific macrophage subpopulations as the analysis of total macrophages might underestimate changes in innate immunity[23]. Eating a HFD shifted colonic macrophages towards more pro-inflammatory subpopulations, which is reminiscent of changes observed in inflammatory bowel disease[10]. The fat source of the diet, but not the fiber content, was a major influence on the magnitude of the innate response in the gut and adipose tissue and in glycemic control. Rather than leading to a specific gut microbiota composition, feeding HFD led to dysbiosis, which differed depending on the dietary fat source. However, gut microbiota seem to be a prerequisite for monocyte recruitment as germ-free mice, being protected from metabolic disease[2], had very low numbers and no increase in pro-inflammatory colonic macrophages upon HFD. This finding suggested that the number of colonic macrophages is linked directly to glycemic control, which was confirmed by a gradual improvement in glucose tolerance upon dose-dependent pharmacological macrophage depletion. To causally link intestinal macrophages and glucose homeostasis, we applied clodronate liposomes intrarectally, which resulted in macrophage depletion in the proximal colon. This localized macrophage depletion allowed us to study the role of colonic macrophages in regulating glucose homeostasis, without confounding effects caused by systemic macrophage depletion or an increase of pro-inflammatory cytokines, as found with systemic clodronate liposome administrations[17]. Intrarectal clodronate liposomes resulted in improved glucose homeostasis similar to the extent of systemic macrophage depletion, thus establishing a causal link between colonic macrophages and glucose homeostasis. In addition, co-culture experiments demonstrated a detrimental effect of colonic macrophages isolated from mice fed a HFD on β-cell function.

To understand the mechanism of intestinal inflammation upon HFD, and thus potentially identify therapeutic targets, we performed single-cell RNA-sequencing of colonic macrophages. We found an interferon signature and a change in mitochondrial metabolism upon HFD, which converge in mTOR signaling as a common regulatory pathway[20,21]. Thus, we postulated that altered mTOR signaling might be involved in the HFD-driven transcriptional response in macrophages. We found that mTOR activity was indeed enhanced in colonic macrophages in HFD mice. mTOR activation has been described as a response to nutrients and is exaggerated in a nutrient-overabundant state[24,25]. In line with that, mTOR inhibition by intrarectal rapamycin treatment attenuated the inflammatory load in the gut by reducing both the number and activation of pro-inflammatory macrophage subpopulation P2 in HFD-fed mice. Consequently, glucose-stimulated insulin secretion was restored when Min6 cells were co-cultured with colonic macrophages from mice that had been treated with intrarectal rapamycin, suggesting improved insulin secretion capacity. A previous study indicated that inhibiting mTOR in the intestine by intraluminal instillations improves glucose homeostasis by lowering glucose production[26]. In contrast, systemic mTOR inhibition has been associated with insulin resistance[22]. One limitation of our study is that our results cannot answer the question of whether the effect between colonic macrophages and β-cell function is direct or indirect. For example, the cross-talk between pro-inflammatory colonic macrophages and β-cells could be direct through migration as monocyte-derived CXCR1[inter] macrophages of the gut have a high migratory potential[27]. Alternatively, cytokines or other secretory products from pro-inflammatory macrophages could disseminate to the pancreas through the blood circulation, lymphatic vessels, or neuronal circuits. A potential neuronal connection between the gut and β-cells could be via enteric-associated neurons,

which have been shown to impact metabolic control[28]. Additionally, we cannot fully exclude a role of GLP-1 as we did not measure local GLP-1 levels (i.e., in portal blood). Besides, also mTOR-independent pathways could play a part in the HFD induced colonic macrophage activation.

When comparing colonic macrophage depletion and mTOR inhibition, rectal rapamycin had more subtle metabolic effects than rectal clodronate, which improved not only β-cell function, but whole-body glucose tolerance and insulin sensitivity (Supplementary Fig. 11). Rectal clodronate liposomes could have been more effective as they specifically target macrophages, while rectal rapamycin might also influence cells other than macrophages in the intestine. However, the strategy to intrarectally inject rapamycin was the most specific method currently available to target mTOR activation of colonic macrophages. Alternatively, we cannot exclude the possibility that a small amount of intrarectally applied rapamycin reached the systemic circulation, counteracting the improvements in β-cell function. However, we found no attenuation of mTOR activity in tissues other than the gut, nor an insulin resistance phenotype in mice treated intrarectally with rapamycin.

Our findings are of high clinical significance as we recently confirmed that obese human subjects also exhibit increased pro-inflammatory colonic macrophages[29]. The validation of our pre-clinical data in human obesity is noteworthy. Usually, there is considerable heterogeneity among human subjects due to their different lifestyles. So far, an increase of pro-inflammatory intestinal macrophages has only been documented in inflammatory bowel disease[10,30]. Intestinal macrophages are known to originate from circulating monocytes[31]. A specific CD14[++]CD16[+] intermediate monocyte subpopulation that shares similar expression pattern with the pro-inflammatory colonic macrophage subpopulation P2 (HLA-DR[+]CD163[-]), is concurrently elevated in obese subjects, which is characterized by a pro-inflammatory phenotype[32] and known to correlate with adiposity and cardiovascular risk[33,34]. Monocytes could thus be activated by dietary intake[35] and, when replenishing intestinal macrophages, alter gut immunity.

The overall aim of this research is to find immune-modulatory treatments that can improve glucose metabolism. Our data emphasize the need for tissue- or even cell-specific treatment strategies, potentially targeting specific pathways involved in colonic macrophage activation upon HFD, such as activated mTOR signaling. Our approach with intrarectal administrations has the advantage that it is non-invasive and potentially transferrable to humans, while at the same time does not affect all macrophage compartments in the body. Targeting colonic macrophages might thus represent a potential therapeutic approach for treating β-cell dysfunction at the onset of metabolic disease.

## Methods

**Mice.** Male C57BL/6 (B6) mice (see Supplementary Data 2) were maintained in our SPF facility at room temperature (22 °C) on a 12 h light/12 h dark cycle. Germ-free B6 mice were bred and maintained in flexible-film isolators or in individually ventilated cages (IVC) at the Clean Mouse Facility, University of Bern, Switzerland. For all high-fat diet (HFD) studies, 5–8 week-old weight-matched mice were fed either a coconut-based HFD (60%; #D12331, Research Diets or Sniff), or a chow diet for up to 3 months. The following additional diets were used to assess the influence of different diet compositions and the microbiota: a purified control diet, which contains nearly no fibers (Starch: 10%; #D12450Ji; Research Diets), a lard-based HFD (60 %; D12492i; Research Diets), an olive oil-based HFD (50%; D12331 mod.; Ssniff) and the corresponding control coconut-based HFD (50%; D12331 mod.; Ssniff). Mice were randomized into different groups according to their starting weights. See diet details in Supplementary Table 1.

**Study approval.** All animal procedures were approved by the local Animal Care and Use Committee and performed in accordance with Swiss Federal regulations.

**Macrophage depletion and mTOR inhibition**. For dose-dependent depletion of macrophages, 4–5 weeks old mice on a HFD were orally treated with 50, 100, or 200 μg/g CSF1R inhibitor (BLZ945; MTA Novartis, Basel, Switzerland) or its vehicle (20% Captisol; Ligand, San Diego, US) for up to ten weeks (5 times/week). Treatment started 1 week before HFD feeding.

To specifically deplete colonic macrophages, clodronate liposomes were injected intrarectally in anesthetized mice with a flexible gavage needle coated with lubricant. The colon-specific depletion was compared to systemic depletion by intraperitoneal application of clodronate liposomes. The clodronate or control PBS liposomes (Liposoma B.V.) were injected every other day (100 μL/500 μg/injection) starting from day −4 until day 6 after the start of HFD feeding. To inhibit mTOR activation, mice were treated intrarectally or intraperitoneally (i.p.) with 3 mg rapamycin/kg/every other day (Cat#HY-10219, MedChemExpress), according to the application of clodronate liposomes.

**Isolation of colonic macrophages**. Intestinal lamina propria lymphocytes were isolated from the colon of mice adapted to the lamina propria isolation protocol of Baranska et al.[36]. For mouse colon samples, the length was measured, fat was removed, the tissue was first cut open longitudinally, then cut into 1 cm pieces and washed in ice-cold DPBS or HBSS (without Mg/Ca). To remove the epithelial layer, tissue pieces and biopsies were washed twice in HBSS/2 mM EDTA shaking 20 min at 37 °C. Afterwards, they were washed twice in HBSS and transferred into a gentle MACS C-tube (#130-096-334, Miltenyi Biotec) containing 3 mL Complete IMDM Medium (1x IMDM, 10% FBS, P/S, Glutamax). Next, 3 mL 2x Collagenase VIII (#C2139, Sigma-Aldrich) digestion solution (Complete IMDM, 2 mg/mL Collagenase VIII, 25 μg/mL DNase I) was added to start enzymatic digestion by shaking at 37 °C (25–30 min). Then, digested tissue was homogenized by using the gentleMACS Octo Dissociator (Militenyi Biotec; program: ms_intestine-01). Later, digestion was stopped by EDTA. Next, leukocytes were enriched by using a percoll gradient (40%/70%, #GE17-0891-01, GE Healthcare) and centrifuged (600 g, 20 min, 22 °C, brake and acceleration 0 or 1). Afterwards, the lymphocyte ring was collected from the interphase, washed (550 g, 5 min, 22 °C) with FACS Buffer (1xDPBS, 0.5% BSA, 5 mM EDTA). Finally, the cells were resuspended in 200 μL FACS Buffer (DPBS/0.5% BSA/5 mM EDTA) containing Fc Blocking and filtered through a 35 μM strainer FACS tube (#352235, Corning).

**Isolation of macrophages in other tissues**. Adipose and liver tissue were minced with scissors and digested by shaking in a Collagenase IV (#C2139, Worthington) solution (1x HBSS, 10 mM HEPES, 1.5 mg/mL Collagenase IV and 8.25 μg/mL DNAse I) for 20–30 min at 37 °C and 400 rpm. Digestion was stopped by adding FACS buffer, and suspension was filtered through cotton gauze. Erythrocytes were removed by using Red Cell Lysis Buffer (154 mM NH4Cl, 10 mM KHCO3, 0.1 mM EDTA). Liver leukocytes were enriched with a 70%/40% percoll gradient. Later, cells were washed and filtered for FACS staining.

Peritoneal macrophages were isolated by lavage of the peritoneum with 10 mL FACS buffer. Erythrocytes were lysed with Red Cell Lysis Buffer, and the remaining cells were washed and filtered for FACS staining. To isolate microglia, whole brains were excised from the skull and mechanically dissociated in FACS buffer by using a Dounce-homogenizer (#D9938-1SET, Merck). Cells were then passed through a 70 μm cell strainer, washed with FACS buffer, and enriched by performing a 70%/37% percoll gradient (30 min, 750 g, minimal brake). The microglia-containing interphase was subsequently collected and filtered, then washed and used for FACS analysis.

**Flow cytometry analysis**. To reduce unspecific binding, the Fc receptor was blocked with CD16/32 prior to incubation with monoclonal antibodies (mAbs) for 30 min[-1] h on ice. All mAbs used for flow cytometry were listed in Supplementary Data 2. See gating for colonic macrophages in Fig. 1a (similar to[8]). Although originally described for skin macrophages[8,37], the same gating strategy can be applied to characterize colonic macrophages[8,37]. In the case of MitoTracker staining, cells were stained with Mito-Tracker probes (Thermo Fisher Scientific) at 37 °C in the dark (10 nM MitoGreen (for mitochondrial mass) and 5 n MitoRed (for mitochondrial membrane potential) for 20 min in 10% FACS Buffer, 1 μM MitoSOX (for reactive oxygen species) for 10 min in 1x HBSS (no Mg/Ca)) after the surface staining. To assess mTOR activation, cells were fixed with BD Fix I Buffer (10 min, 37 °C) after surface staining. Subsequently, cells were washed and permeabilized by adding BD Perm Buffer II (30 min on ice), followed by staining for 1 h at RT with anti-pS6 and anti-pS473. Samples were acquired with a BD LSRIIFortessa (BD), and analyzed with FlowJo software 10.6.1 (BD).

**Intracellular flow cytometry (ICFC)**. Colonic lamina propria cells were isolated as described above (see isolation of colonic macrophages). To enhance intracellular cytokine staining all buffers (1xHBSS, 1xHBSS/EDTA), a digestion solution contained the protein transport inhibitors 5 μg/mL Brefeldin A (Sigma, #B6542) and 1 μg/mL Monensin (Sigma, # M5273). After percoll gradient, cell pellet was split into two tissue culture tubes (Corning PYREX # 99445-10) and incubated in 450 μL Complete IMDM Medium without or with 1 μg/mL Lipopolysaccharides (LPS) (Sigma-Aldrich, LPS E. coli 0111: B4) for 4.5 h at 37 °C in the incubator. Afterwards, cell suspension was transferred into a deep well plate, washed twice with FACS buffer, and later stained with surface antibodies (50 μL antibody mix/sample)

for 30 min at 4 °C. Then cells were washed with FACS buffer, resuspended in IC Fixation buffer (150 μL/well) (eBioscience Intracellular Fixation & Permeabilization Buffer #88-8824-00), and incubated for 30 min at RT. Later, samples were washed with 1x Perm buffer (150 μL/well). Intracellular antibodies were added in 1x Perm Buffer (50 μL/sample), and staining was performed 45–60 min at RT. Staining was stopped by washing with FACS buffer, and cells resuspended in FACS Buffer were used for further flow cytometry analysis.

**Metabolic assessments**. Metabolic phenotype was assessed by glucose and insulin tolerance tests (GTT/ITT) performed at 1, 4, and 12 weeks of HFD feeding. For a GTT, mice fasted 6 h. After intraperitoneal (IPGTT) or oral (OGTT) injection of glucose (2 g/kg body weight) blood glucose was monitored from the tail vein after 15, 30, 60, 90, and 120 min by using a glucometer (Freestyle, Abbot). For active GLP-1 measurements, 25 mg/kg sitagliptin (Cat#sc-364620, Santa Cruz) was injected i.p. 30 min before oral glucose application. To block GLP-1 or parasympaticus action, 236 μg/kg exendin (9-39) (#H-8740, Bachem) or 5 mg/kg atropine (#A0257-5G, Sigma), respectively, were i.p. injected at timepoint −30 min prior glucose application. ITT was performed after 3 h of fasting by injecting 1–2 U/kg body weight insulin i.p. (Actrapid HM Penfill, Novo Nordisk). Glucose levels were measured at 0, 15, 30, 60, 90, and 120 min after injection.

Plasma insulin, GLP-1, TNF, and IL-6 were quantified by electrochemiluminescence (MESO SECTOR S 600) by using kits from Meso Scale Diagnostics (MSD, Rockville, MD, USA), according to the manufacturer's instructions: Mouse/Rat Insulin Kit (#K152BZC), V-PLEX Plus Proinflammatory Panel 1 Mouse Kit (#K15048G), V-PLEX GLP-1 Active Kit vers. 2 (#K15030D).

**Quantitative RT-PCR analysis**. RNA was extracted from distal colon and epididymal adipose tissue with the NucleoSpin RNA kit (#740955.250, Macherey-Nagel) or with the RNeasy Plus Universal Mini kit (#73404, QIAGEN) according to each manufacturer's instructions. Reverse transcription was performed with GoScript™ Reverse Transcription Mix (#A2801, Promega). For qPCR, GoTaq qPCR Master Mix (#A6002, Promega) on a ViiA7 Real-Time PCR System (Thermo Fisher Scientific) was used. Primer sequences (Microsynth, Balgach, Switzerland) are listed in Supplementary Table 2. The data presented correspond to the mean of $2^{-\Delta\Delta Ct}$ after being normalized to housekeeping genes B2m and Ppia.

**Isolation of pancreatic islets**. Pancreatic mouse islets were isolated by injecting collagenase IV (1.4 mg/mL; Worthington) digestion solution into the pancreas via the common bile duct. The perfused pancreas was digested at 37 °C for 30 min, then quenched (1x HBSS, 1 M HEPES, 0.5% BSA) and filtered. Islets were hand-picked under a stereoscopic microscope and cultured in RPMI-1640 (containing 11.1 mM glucose, 10% FBS, 100 U/mL P/S, 2 mM Glutamax, 50 μg/mL Gentamycin, 10 μg/mL Fungison). For flow cytometry analysis, islets were washed in PBS/0.5 mM EDTA and trypsinized. About 100 islets/tube were collected.

**Glucose-stimulated insulin secretion (GSIS)**. For GSIS, we handpicked primary mouse islets or plated Min6 cells (25,000 Min6 cells and 5,000 sorted colonic macrophages per well) in a 24-well plate, which were incubated in RPMI-1640 medium overnight (16 h). The next day, islets or Min6 cells were washed and pre-incubated in Krebs-Ringer-bicarbonate buffer (KRB: 115 mM NaCl, 4.7 mM KCl, 2.6 mM CaCl2 2H2O, 1.2 mM KH2PO4, 1.2 mM MgSO4 2H2O, 10 mM HEPES, 0.5 % BSA, pH 7.4) containing 2.8 mM glucose. After 1.5 h (islets) or 2 h (Min6), the buffer was replaced with KRB containing low (2.8 mM, basal) or high (16.7 mM, stimulated) glucose, and the supernatant was collected after 1 h to assess basal and glucose-stimulated insulin release. The stimulatory index was defined as the ratio of stimulated insulin secretion at 16.7 mM/h to basal insulin secretion at 2.8 mM/h. To obtain insulin content, islets or Min6 cells were resuspended in 0.18 mol/l HCl in 70% ethanol and incubated at least 1 h at −20 °C. Secreted and content insulin was measured with the Mouse/Rat Insulin Kit (#K152BZCMeso Scale Discovery).

**Microbiota analysis**. For genomic DNA extraction from stool samples, contents from cecum and colon, or feces, were frozen in 2 mL tubes and stored at −80 °C until extraction. To extract genomic DNA from feces, the QIAamp FAST DNA Stool Mini Kit (#51604 Qiagen, Hilden, Germany) was used following the vendor's instructions, except for the following adjustment: homogenization of stool particles was performed with 100 mg baked glass beads (Sigma Aldrich) by using a tissue lyser for 3 min, 30 Hz per run (Retsch MM400). DNA concentration was measured by Nanodrop2000 (Thermo Fisher Scientific).

For 16S amplicon PCR, 100 ng of bacterial DNA were used to amplify the V5/V6 region of the 16S ribosomal gene by PCR by using Platinum Taq DNA polymerase (Invitrogen). We used barcoded forward fusion primers 5′- CCATCTCATCCCTG CGTGTCTCCGACTCAG-BARCODE-ATTAGATACCCYGGTAGTCC-3′, where core primers have been modified by adding a PGM sequencing adaptor, a GT-spacer, and unique barcode (see Supplementary Table 3), that allows us to pool up to 96 different barcodes in combination with the reverse fusion primer 5′-CCTCTCT ATGGGCAGTCGGTGATACGAGC-TGACGACARCCATG-3′[38–40]. All primers were used at a 10 μM working concentration. The following were the cycling conditions: initial 5 min denaturation at 94 °C, followed by 35 cycles of 1 min denaturation at 94 °C, 20 s annealing at 46 °C, and 30 s extension at 72 °C. The final

extension step took place for 7 min at 72 °C. The PCR product (~350 bp) was loaded on a 1% agarose gel, cut out with a scalpel, and extracted by using the QIAqick Gel Extraction Kit protocol (#28706, Qiagen). The resulting dsDNA concentration was measured by Qubit dsDNA HS Assay Kit (#Q32854, Thermo Fisher Scientific).

For 16S sequencing, up to 96 libraries were diluted at 26 pM and were pooled. Libraries were prepared with the OT$^2$ HiQ View 400 kit, and emulsion PCR was performed on the Ion OneTouch 2 (OT$^2$) instrument (ThermoFischer). The template-positive Ion Sphere Particles containing clonally amplified DNA were enriched with the Ion OneTouch ES instruments (ThermoFisher). Sequencing was carried out by using the IonPGM HiQ View Sequencing 400 Kit with the Ion Personal Genome Machine (PGM) System on an Ion 316 Chip v2 (ThermoFisher)[41].

To analyze 16S data, samples with fewer than 1,000 reads were excluded from the analysis if not stated otherwise. Data analysis was performed by using the QIIME pipeline version 1.8.0[42]. OTUs were picked at a threshold of 97 % similarity by using usearch61_ref v.6.1.544[43], followed by rarefaction and taxonomy assignment with the GreenGenes database (greengenes.secondgenome.com). Multivariate analysis by linear models (MaAsLin) in the R package was used to find associations between genotype and microbial community abundance[44].

**RNA-sequencing**. For single-cell RNA-sequencing (scRNA-seq) (liveCD45$^+$Lin$^-$CD11b$^+$CD24$^-$ and CD64$^+$ or Ly6C$^+$) CD11b$^+$nonDCs colonic macrophages (see gating strategy[8] and Fig. 1a) were sorted from mice fed 1 week of HFD ($n = 2$) or chow diet ($n = 2$) by using FACS Aria III (BD Biosciences). Cell suspensions were loaded into the wells of a 10X Genomics Chromium Single Cell Controller (one well per mouse replicate). Single-cell capture and cDNA and library preparation were performed with a Single Cell 3′ v2 Reagent Kit (10X Genomics) according to the manufacturer's instructions. Sequencing was performed on one lane of an Illumina NexSeq 500 machine flow cell at the ETH Zurich Genomics Facility in Basel, Switzerland.

Data were analyzed by the Bioinformatics Core Facility, Department of Biomedicine, University of Basel, Switzerland. Read quality was assessed with the FastQC tool (version 0.11.5). In brief, sequencing files were processed with Kallisto (version 0.46.0) and BUStools (version 0.39.2) to perform sample and cell demultiplexing, read pseudo-alignment to the mouse transcriptome (Ensembl release 97), and to generate a UMI counts table[45,46]. Further processing of the UMI counts table was performed by using R 3.6.0 and Bioconductor 3.10 packages[47], notably DropletUtils (version 1.6.1)[48,49], scran (version 1.14.5), and scater (version 1.14.5)[50], following mostly the steps illustrated in the simpleSingleCell Bioconductor workflow[51].

Based on the distributions observed across cells, cells with library sizes lower than 795, total number of features detected lower than 317, or with a fraction of UMI counts attributed to the mitochondrial genes of 0% or higher than 7% were filtered out[52]. Low-abundance genes with average normalized log$_2$ counts lower than 0.003 were filtered out. This resulted in a filtered matrix including UMI counts for 11,820 genes and 5797 cells (3013 from chow-fed mice and 2784 from HFD-fed mice). UMI counts were normalized with size factors estimated from pools of cells to deal with dominance of zeros in the matrix[51,53]. A mean-dependent trend was fitted to the variances of the log expression values of endogenous genes to distinguish between genuine biological variability and technical noise (*trendVar* function of the scran package with loess trend and span of 0.05)[54]. The fitted trend was used to subtract technical noise from the data by using the *denoisePCA* function, retaining the 8 first principal components of the PCA for later analysis.

The package SingleR (version 1.0.0) was used for reference-based annotation of the cells and identification of likely contaminants in our dataset[55]. We used the Immunological Genome Project (ImmGen) mouse microarray dataset[18] as reference, and eliminated 377 cells not annotated to the broad cell types macrophages or monocytes.

Clustering of cells was done on normalized log-count values by using hierarchical clustering on the Euclidean distances between cells (with Ward's criterion to minimize the total variance within each cluster; package cluster version 2.1.0). The 6 clusters used for following analyses were identified by applying a dynamic tree cut (package dynamicTreeCut, version 1.63-1).

Differential expression between HFD and chow conditions, stratified by cluster, was performed by using a pseudo-bulk approach[56]: UMI counts of cells from each sample in each cluster were summed. This resulted in 4 samples per cluster, aggregated form of 29 to 776 cells. Cluster P2.2 was excluded from the analysis because it contained too few chow cells. For each cluster, we only retained genes with CPM (normalized counts per million mapped reads) values above 1 in at least 2 of the 4 pseudo-bulk samples, and detected in at least 5 % of the individual cells.

The package edgeR (version 3.28)[57] was used to perform TMM normalization[58] and to test for differential expression with the Generalized Linear Model (GLM) framework. Genes with a false discovery rate (FDR) lower than 1 % were considered to be differentially expressed. Gene set enrichment analysis (GSEA) was performed with the function camera[59] by using the default parameter value of 0.01 for the correlations of genes within gene sets, on gene sets from the hallmark collectionsupp[60] of the Molecular Signature Database (MSigDB, version 7.0)[61], or on DoRothEA v2 regulons[62]: human TOP10score regulons were downloaded from https://github.com/saezlab/DoRothEA and we obtained the corresponding mouse regulons by considering 1-to-many orthologs of the human genes in each regulon (by using

Ensembl Compara release 97). We tested only gene sets containing at least 5 genes from the filtered dataset, and considered significant those with a FDR lower than 10%.

**Statistics and reproducibility**. The data are presented as mean ± standard error of the mean (SEM), with the numbers (n) of experiments and mice indicated in the Fig. legends. To test the statistical difference between two groups, an unpaired Mann–Whitney $U$ test with two-tailed distribution was run with Prism8 software (GraphPad Software, San Diego, CA). Two-sided p-values of 0.05 or less were considered to be statistically significant.

**Reporting summary**. Further information on research design is available in Nature Research Summary linked to this article.

## Data availability
The source data is available within the paper in Supplementary Data 1. Other data supporting the findings of this study are available from the corresponding author upon request. Transcriptome sequencing data of colonic macrophages have been deposited in the Gene Expression Omnibus, with the accession number GSE143351. Datasets generated during this study are available by entering token **unoveyewbnyzpuj** into the search box.

## Code availability
All major software and code used to analyze the datasets described in this paper are referenced above.

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

## Acknowledgements

The authors thank Marianne Böni-Schnetzler and Marc Donath for their constant support by sharing their laboratory space and expertise; Marc Stawiski, Neena Parayil, Friederike Schulze, Erez Dror, Anna Steinert, Katarina Radulovic and Cristina Kalbermatter for technical contributions, and Romano Schneider for constructive feedback. Danny Labes for his flow cytometry expertise; Nicole Kirchhammer for help with intracellular staining; Glenn Bantug and Mohammedyaseen Syedbasha for feedback on mTOR and metabolic assessments; Anna Baranska for sharing her expertise and isolation protocol in murine intestinal macrophages; and the members of the Donath, Hess, Recher, Berger and Mehling laboratories at the Department of Biomedicine of the University Hospital of Basel for advice and feedback. This study was supported by grants from the Swiss National Science Foundation (PZ00P3_161135 to CCW), the Goldschmidt-Jacobson Foundation, the Jubiläumsstiftung Swiss Life, the Olga Mayenfisch Foundation, the Basler Diabetesgesellschaft (to C.C.-W.), the Nikolaus and Bertha Burckhardt-Bürgin-Stiftung (to T.R.) and from the Research Fund for Excellent Junior Researchers of the University of Basel (to T.R.). S.G.V. was funded through the Peter Hans Hofschneider Professorship provided by the Stiftung Molekulare Biomedizin.

## Author contributions

Experimental design: T.R., D.M., C.C.-W. Experimental execution: T.R., L.K., S.A.A., A.B., Z.B., S.J.W., D.M., A.T., J.W., L.R., C.M., N.F.T., S.G.V., C.C.-W. Data analyses: T.R., L.K., S.A.A., J.R., B.Y., S.G.V., C.C.W. Figure preparation, manuscript writing: T.R., C.C.-W. Editing: T.R., L.K., S.J.W., D.M., A.B., J.R., E.D., Z.B., S.A.A., S.G.V., B.Y., C.M., N.F.T., A.T., J.W., A.J., S.W., D.K., J.N., S.H., C.C.-W. C.C.-W. is the guarantor of this work.

## Competing interests

The authors declare no competing interests.
