## [Peer Review File · Communications Biology]

Reviewers' comments:

Reviewer #1 (Remarks to the Author):

The manuscript evaluates the role of colonic macrophage subpopulations in glucose metabolism. The authors observe that colonic macrophages increased in mice fed HFD. Colonic macrophage activation upon HFD was characterized by an interferon response and a change in mitochondrial metabolism, which converge in mTOR as a common regulator. The results are comprehensive and generally support the authors' conclusions. These findings may benefit from some additional clarification, as detailed below.

Comments

- What is the translational relevance of this study? Did the authors characterize monocytes isolated from obese subjects?
- The authors demonstrated that the source of dietary fat modulated the pro-inflammatory innate immune response and glycemic control. Olive oil-based HFD should be added to the study.
- Line 164-166: "the lack of gut microbiota in germ-free mice prevented them from recruiting and hence increasing pro-inflammatory colonic macrophages upon HFD". It could be very interesting to evaluate the effects of probiotic administration on gut microbiota composition and colonic macrophages upon HFD.
- The manuscript should be copy edited by a native English speaker to correct minor grammatical and contextual errors.

Reviewer #2 (Remarks to the Author):

This study examined the role of colonic macrophages in insulin resistance that develops in high fat diet fed mice. The authors nicely show that accumulation of inflammatory macrophages in the colon occurs early after HFD initiation, prior to macrophage accumulation in the adipose tissue, in a microbiome dependent fashion. Depletion of systemic and/or colonic macrophages improves glucose metabolism. The data described occupy the first half of the manuscript and are strong, well designed, and clearly presented.

I find the second half of the manuscript, where the authors attempt to implicate mTOR pathways as a mechanism by which colonic macrophages influence glucose metabolism, weaker and with potential methodologic and interpretative flaws. The authors find that in vitro augmentation of islet cell insulin secretion by colonic macrophages correlates with their effects in vivo. Using scRNA Seq, they identify a potential role for mTOR mediated signaling in affecting islet secretion. However, ultimately the authors do not show a causative role for mTOR signaling in mediating the glycemic effects of colonic macrophages.

Nevertheless, I think this is an interesting and novel study that merits publication pending revisions.

Comments:

1. Please clarify which HFD was used for the experiments in Figure 1, given the several different diets used subsequently in the paper.
2. How do the authors interpret the differential effect of the lard- and coconut- based HFD on anti-inflammatory CCR2- macrophage subset? Could not the CCR2- changes contribute to the differences in glycemic control seem between these diets?
3. For the intra-rectal clodronate-treated mice, did the authors examine the effect of the treatment on macrophage numbers in the spleen, small bowel, and/or peritoneum? Peritoneal macrophages in particular may be exposed to the intra-rectal clodronate and potentially mediate some of the effects on systemic glucose metabolism observed.

4. I am not sure I agree with the authors interpretation of the exendin inhibitor experiment in the clodronate treated mice. It seems that a substantial proportion of clodronate's beneficial impact on glucose metabolism are reversed with the GLP-1 antagonist. Although the measured systemic GLP-1 levels are not different, did the authors examine portal GLP-1 levels, which more directly influence islet function? Moreover, the degree of GLP-1 increase the authors observe is quite low, especially given these mice were pre-treated with a DPP-IV inhibitor. Did the authors consider measuring GLP-1 following an oral glucose load in these two treatment groups?
5. For the in vitro cytokine data following mTOR inhibition, what is the rationale for the cytokine panel chosen? In particular, the scRNA Seq data suggested an IFN- α and IFN- γ signature in the colonic macrophages – why were these cytokines not measured?
6. What happens to insulin secretion in the rectal rapamycin treated animals? Based on the hypothesis, mTOR inhibition should lead to augmented insulin secretion?
7. Given the lack of effect of rectal rapamycin on systemic glucose tolerance and the fairly mild impact on the in vitro surrogate of islet insulin secretion, isn't an alternative possibility that colonic macrophages act through a mTOR independent pathway? If authors agree, I would consider adding discussion to this point.
8. The authors indicate that their co-culture experiments with Min6 cells show a “deleterious effect of colonic macrophage-derived cytokines.” In these co-culture experiments, how can the authors rule out a direct cell mediated effects? It seems reproducing augmented insulin secretion from Min6 cells with conditioned medium from colonic macrophages is necessary to implicate a soluble factor, e.g. cytokines.
9. Did the authors measure signs of systemic inflammation (serum cytokine levels, e.g TNF & IL-6, adipose tissue inflammation) in the colonic macrophage depleted animals? Does colonic macrophage depletion impact barrier function? This could be addressed by measuring serum LPS levels, or alternatively, by directly measuring gut permeability with gavage of labeled dextran particles.

9th January, 2022

Manuscript COMMSBIO-21-2952-T– Responses to Reviews

We thank the editor and the reviewers for their thoughtful comments and valuable input regarding our manuscript entitled **“Targeting colonic macrophages improves glycemic control in high-fat diet-induced obesity”**. Please find below a point-by-point response to all issues raised.

Point-by-point response to the reviewers' comments

Reviewer #1 (Remarks to the Author):

The manuscript evaluates the role of colonic macrophage subpopulations in glucose metabolism. The authors observe that colonic macrophages increased in mice fed HFD. Colonic macrophage activation upon HFD was characterized by an interferon response and a change in mitochondrial metabolism, which converge in mTOR as a common regulator. The results are comprehensive and generally support the authors' conclusions. These findings may benefit from some additional clarification, as detailed below.

Comments

- What is the translational relevance of this study? Did the authors characterize monocytes isolated from obese subjects?

Response:

Originally, our manuscript contained human data confirming increased inflammatory colonic macrophages (subpopulation P2) in obese compared to non-obese subjects (Fig. 1 of the letter). These biopsies came from patients undergoing screening colonoscopies for colorectal cancer screening. In the meantime, we published a detailed and comprehensive analysis on human gut biopsies from several locations along the gastrointestinal tract of obese and non-obese subjects, confirming the shift towards more inflammatory P2 macrophages not only in the colon, but also at other locations of the gastrointestinal tract (stomach, duodenum, colon) (1). We decided to remove the data on human colon macrophages from the current Communications biology manuscript in order not to report this piece of data twice.

The fact that we were able to confirm our data in human biopsy samples stresses the clinical importance of our findings. Intermediate blood monocytes were also increased in obese humans, suggesting enhanced recruitment to the gut. Moreover, unhealthy lifestyle habits such as low vegetable intake and high processed meat

consumption correlated with increased inflammatory macrophages in the gut, indicating potential triggering factors of altered gut innate immunity in an obese condition (1). We refer to our paper on page 16 in the discussion.

a Human colonic macrophages

b

c Colon

d Peripheral blood mononuclear cells

Fig. 1. Pro-inflammatory colonic macrophages are increased in human obese subjects. a, Flow cytometry strategy of human colonic macrophages (c-Macs: CD14^{high} (pro-inflammatory P1, P2 and intermediate P3) and CD14^{low} (anti-inflammatory/resident P4, P5 subpopulations).

b, c, Proportional distribution (b) and frequencies of human c-Macs (total, CD14^{high}/CD14^{low} and P1-P5 subpopulations) (c) from colon biopsies of lean (BMI < 27 kg/m², n=13) or obese subjects (BMI > 32 kg/m², n=8). d, Frequencies of blood monocytes and their subpopulations from lean (n=21) or obese subjects (n=15). One data point represents one subject. Statistical data are expressed as mean±SEM. *p<0.05. **p<0.01, ***p<0.001, unpaired Mann-Whitney U test with two tailed distribution.

- The authors demonstrated that the source of dietary fat modulated the pro-inflammatory innate immune response and glycemic control. Olive oil-based HFD should be added to the study.

Response:

We have now assessed colonic macrophage subpopulations of mice fed 1 week an olive oil-based HFD compared to the respective control diets. One caveat is that olive-based diet can only be produced with a maximum fat content of 50 kcal % as opposed to 60 kcal % in the other HFDs used in this study (coconut-based and lard-based) as a liquid oil such as olive oil cannot be pelleted with a higher kcal proportion as it would otherwise leak out. As a control diet for these new experiments, we used coconut-based HFD with a fat content of 50 kcal % as well as a control chow. The details of all diets used in our study are listed in the Extended Data Table 2.

We found that mice fed one week an olive-oil based HFD showed impaired glucose tolerance, but a less extreme increase in P2 colonic macrophages compared to mice fed a coconut-based HFD. The detailed data on the metabolic phenotype; the macroscopic measurements of stomach, caecum and colon; colonic macrophages and adipose tissue macrophages are shown below. We have now mentioned these data in the results on page 7.

Fig. 2. Effects of one-week olive oil HFD feeding on the metabolic phenotype and macrophage subpopulations in colon and adipose tissue. Wild-type mice were fed 1 week (wk) 50 kcal % olive oil-based HFD (orange) or 50 kcal % coconut-based HFD (teal): **a**, Intraperitoneal glucose tolerance test (IPGTT), insulin and body weight after 1 wk of HFD (Chow/Coconut/Olive oil n=6/6/6). **b**, Macroscopic parameters of colon length, caecum weight and stomach weight after 1 wk of HFD (Chow/Coconut/Olive oil n=6/6/6). **c**, Frequency (%) and absolute numbers (#) of colonic macrophages (c-Macs) (Chow/Coconut/Olive oil n=5/6/6). **d**, Frequency (%) and absolute numbers (#) of adipose tissue macrophages (ATMs) (Chow/Coconut/Olive oil n=6/6/6) after 1 wk of HFD. Statistical data are expressed as mean±SEM. Data are representative of two experiments, with each data point representing one individual mouse. *p<0.05, **p<0.01, unpaired Mann-Whitney U test with two tailed distribution.

- Line 164-166: “the lack of gut microbiota in germ-free mice prevented them from recruiting and hence increasing pro-inflammatory colonic macrophages upon HFD”. It could be very interesting to evaluate the effects of probiotic administration on gut microbiota composition and colonic macrophages upon HFD.

Response:

We thank the reviewer for this interesting point. We originally used antibiotic treatments to address the role of gut microbiota in HFD-induced changes of colonic macrophages. However, we realized that antibiotic treatment only dampens the microbiota load, such that we used germ-free mice to address the role of gut microbiota in HFD-induced changes of colonic macrophages (see Fig. 2e-f, Extended Data Fig. 4).

The follow-up question regarding the effect of probiotics on gut microbiota and colonic macrophage subpopulations is very interesting, however, we feel this is out of the scope of the current study and should therefore be addressed in a follow-up study.

- The manuscript should be copy edited by a native English speaker to correct minor grammatical and contextual errors.

Response:

The paper has been edited by a professional English language proofreading and editing service (see track change mode of the manuscript).

Reviewer #2 (Remarks to the Author):

This study examined the role of colonic macrophages in insulin resistance that develops in high fat diet fed mice. The authors nicely show that accumulation of inflammatory macrophages in the colon occurs early after HFD initiation, prior to macrophage accumulation in the adipose tissue, in a microbiome dependent fashion. Depletion of systemic and/or colonic macrophages improves glucose metabolism. The data described occupy the first half of the manuscript and are strong, well designed, and clearly presented.

I find the second half of the manuscript, where the authors attempt to implicate mTOR pathways as a mechanism by which colonic macrophages influence glucose metabolism, weaker and with potential methodologic and interpretative flaws. The authors find that in vitro augmentation of islet cell insulin secretion by colonic macrophages correlates with their effects in vivo. Using scRNA Seq, they identify a potential role for mTOR mediated signaling in affecting islet secretion. However, ultimately the authors do not show a causative role for mTOR signaling in mediating the glycemic effects of colonic macrophages.

Nevertheless, I think this is an interesting and novel study that merits publication pending revisions.

Comments:

1. Please clarify which HFD was used for the experiments in Figure 1, given the several different diets used subsequently in the paper.

Response:

For all high-fat diet (HFD) experiments, mice were fed either a coconut-based high-fat diet (60 % HFD #D12331) or a chow diet for up to 3 months. Additional purified diets with a distinct fat source (coconut versus lard) or without fibers (control starch diet) were used in Fig. 2. This is described in the methods on page 30 of the manuscript.

In the results section, we now added the type of HFD ("coconut-based") also on pages 5, 8, and 9.

Additionally, we made sure we mention the type of HFD ("coconut-based") in the figure legend of each figure.

2. How do the authors interpret the differential effect of the lard- and coconut- based HFD on anti-inflammatory CCR2⁻ macrophage subset? Could not the CCR2⁻ changes contribute to the differences in glycemic control seem between these diets?

Response:

In mice fed a coconut-based HFD, we found increased pro-inflammatory P1 and P2 subpopulations in the colon (Fig. 2b, Extended Data Fig. 3b). In mice fed a lard-based HFD, additionally total intestinal macrophages were increased (Fig. 2b, Extended Data Fig. 3b), indicating an enhanced innate immune response. In terms of CCR2⁻ anti-inflammatory macrophages, P4 is only a very small cell population (see Fig. 1a of the manuscript) and the main anti-inflammatory subpopulation P5 was not altered at all (Fig. 2b, Extended Data Fig. 3b). Therefore, it seems very unlikely that CCR2⁻ colonic macrophages could have caused the more extreme metabolic phenotype in mice fed a lard-based HFD.

3. For the intra-rectal clodronate-treated mice, did the authors examine the effect of the treatment on macrophage numbers in the spleen, small bowel, and/or peritoneum? Peritoneal macrophages in particular may be exposed to the intra-rectal clodronate and potentially mediate some of the effects on systemic glucose metabolism observed.

Response:

Peritoneal macrophages: In the mice receiving clodronate liposomes intrarectally, we found no significant reduction in the peritoneal macrophages compared to control mice receiving PBS liposomes (Fig. 3, top panel). In contrast, there was a significant reduction of large peritoneal macrophages in mice intraperitoneally injected with clodronate liposomes (Fig. 3, bottom panel). Interestingly, when PBS liposomes (not containing clodronate) were injected intraperitoneally, large peritoneal macrophages were also reduced compared to PBS controls, albeit not as pronounced as with clodronate liposomes. Due to the high variance of data points (most likely due to the method of cell harvest, namely intraperitoneal lavage), we decided to omit this data in the paper.

Fig. 3. Peritoneal macrophages upon rectal or intraperitoneal injection of PBS (red), PBS liposomes (blue) or clodronate liposomes (green) in mice fed a HFD. Each data point represents an individual mouse, LPMs= large peritoneal macrophages, SPM= small peritoneal macrophages, LPM+SPM= peritoneal macrophages (total); *** $p < 0.001$, unpaired Mann-Whitney U test with two tailed distribution, ns= not significant.

Different sections of the intestinal tract:

Proximal colon: The reported depletion of colon macrophages was found in the proximal colon as specified in the results on page 9, the discussion on page 14, and the figure legends on page 25 (Fig. 4a of the letter; see also Fig. 4c of the manuscript).

Distal colon: In the distal part of the colon, there were no significant reductions in colon macrophages upon intrarectal clodronate liposome injections (Fig. 4b).

Small intestine: We did not perform flow cytometry of intestinal macrophages from the small intestine. However, we have performed gene expression analysis from the distal small intestine (distal ileum) of mice receiving clodronate liposomes intrarectally and controls receiving PBS liposomes. We found a small, but significant reduction of the macrophage marker F4/80 in the small intestine, but not CD68 and or cytokines IL-6, IL-1b and IL-10 in mice treated intrarectally with clodronate liposomes compared to mice treated with PBS liposomes (Fig. 4d).

Interpretation: We think that by intrarectal clodronate liposomes, macrophage depletion is maximal at the location where the tip of the flexible needle reaches, hence the proximal colon (Fig. 4c). From there, there seems to be a gradual “dilution” of the macrophage depletion. We specified in the results on page 9 the location of the macrophage depletion and added in the discussion on page 14 that the depletion we achieved was maximal in the proximal colon.

Fig. 4. Peritoneal macrophages upon rectal or intraperitoneal injection of PBS (red), PBS liposomes (blue) or clodronate liposomes (green) in mice fed a HFD. a, proximal colon, b, distal colon, c, tip of the injection needle reaching the proximal colon. d, gene expression of macrophage markers and cytokines in the distal small intestine. Each data point represents an individual mouse, LPMs= large peritoneal macrophages, SPM= small peritoneal macrophages, LPM+SPM= peritoneal macrophages (total);

*** $p < 0.001$, ** $p < 0.01$, unpaired Mann-Whitney U test with two tailed distribution, ns= not significant.

Spleen: Unfortunately, we did not harvest the spleen of these animals.

4. I am not sure I agree with the authors interpretation of the exendin inhibitor experiment in the clodronate treated mice. It seems that a substantial proportion of clodronate's beneficial impact on glucose metabolism are reversed with the GLP-1 antagonist. Although the measured systemic GLP-1 levels are not different, did the authors examine portal GLP-1 levels, which more directly influence islet function? Moreover, the degree of GLP-1 increase the authors observe is quite low, especially given these mice were pre-treated with a DPP-IV inhibitor. Did the authors consider measuring GLP-1 following an oral glucose load in these two treatment groups?

Response:

We agree with the reviewer that the GLP-1 antagonist partially reversed the beneficial impact of clodronate liposomes on glucose metabolism. However, as we could not measure differences in systemic GLP-1 levels among the different groups, we concluded that GLP-1 was not a major contributing factor. However, we cannot fully exclude a local action of GLP-1 as we did not measure portal GLP-1 levels. We mention this limitation of our study in the discussion on page 15.

We measured GLP-1 levels following an oral glucose bolus (Extended Data Fig. 7f and methods on page 34). When comparing our levels with the current literature, we report similar GLP-1 excursions, see also the following references: Fig. 3C in reference (2) or Fig. 5C in reference (3).

5. For the in vitro cytokine data following mTOR inhibition, what is the rationale for the cytokine panel chosen? In particular, the scRNA Seq data suggested an IFN- γ and IFN- α signature in the colonic macrophages, why were these cytokines not measured?

Response:

Interferon measurement: It is important to note that the changes observed in our RNA-Seq data refer to an up-regulated interferon response, rather than to an elevated secretion of interferon.

However, we also measured IFN γ by intracellular flow cytometry in colonic macrophages isolated from mice fed HFD compared to chow fed controls (Fig. 5). We did not find increased interferon levels; however, this is not in contradiction with an increased interferon response.

a

Fig. 5. Comparable IFN- γ production in colonic macrophages of mice fed chow or HFD.

Cytokine panel: The cytokines were chosen based on functional characterization of intestinal macrophages published by Bain et al., 2013 (4). As intestinal macrophages are so few in numbers, the methodology to characterize them in detail is challenging and therefore intra-cellular staining by flow cytometry of a few candidate cytokines (in addition to the markers to identify the different populations) was the best method available up to that time.

6. What happens to insulin secretion in the rectal rapamycin treated animals? Based on the hypothesis, mTOR inhibition should lead to augmented insulin secretion?

Response:

Indeed, we found increased insulin levels upon local mTOR inhibition by intrarectal rapamycin treatment, especially at time points 15 and 30 minutes of the GTT. We have added this data in Fig. 6f and referred to these results on page 12.

7. Given the lack of effect of rectal rapamycin on systemic glucose tolerance and the fairly mild impact on the in vitro surrogate of islet insulin secretion, isn't an alternative possibility that colonic macrophages act through a mTOR independent pathway? If authors agree, I would consider adding discussion to this point.

Response:

We agree and now mention on page 15 in the discussion that also an mTOR independent pathway could be involved.

8. The authors indicate that their co-culture experiments with Min6 cells show a "deleterious effect of colonic

macrophage-derived cytokines.” In these co-culture experiments, how can the authors rule out a direct cell mediated effects? It seems reproducing augmented insulin secretion from Min6 cells with conditioned medium from colonic macrophages is necessary to implicate a soluble factor, e.g. cytokines.

Response:

We agree that our results cannot answer the question whether the effect between colonic macrophages and β -cell function is direct or indirect, i.e. through secreted factors such as cytokines. However, as the yield of colon macrophages is very limited and therefore a high number of mice would be needed to reach discernible levels of secretory factors from colon macrophages *ex vivo*, we have not performed experiments with conditioned medium. Instead, we have adjusted the wording in the results on page 13 and described on page 15 of the discussion that the crosstalk between colonic macrophages and β -cells could be either direct (i.e. through cell migration) or indirect (i.e. secreted factors such as cytokines).

9. Did the authors measure signs of systemic inflammation (serum cytokine levels, e.g TNF & IL-6, adipose tissue inflammation) in the colonic macrophage depleted animals?

Does colonic macrophage depletion impact barrier function? This could be addressed by measuring serum LPS levels, or alternatively, by directly measuring gut permeability with gavage of labeled dextran particles.

Response:

Systemic inflammation: We have extensively characterized the overall “inflammatory” phenotype of clodronate treated and control mice. The readouts include:

- TNF and IL6 => see Extended Data Fig. 6e.
- Macrophage populations in ATMs => see Extended Data Fig. 6b,
- Macrophage populations in liver => see Extended Data Fig. 6c
- Macrophage populations in islets => see Extended Data Fig. 6c
- Macrophage populations in brain => see Extended Data Fig. 6c

Barrier function:

At the time of the experiments, we did not measure gut permeability by labeled dextran particles. Due to preanalytical problems, we were cautious to measure LPS from our stored serum samples. However, we measured different markers of barrier function by gene expression analysis from distal colon, proximal colon and distal small intestine. Thereby, gene expression was not altered in the colon (neither proximal nor distal) of mice on HFD intrarectally treated with clodronate liposomes or control liposomes. In the distal small intestine, there was a slight decrease of tight junction marker 1 (Tjp1) in clodronate liposome treated mice. As this data is very preliminary, we decided not to include it in the manuscript.

Fig. 6. Fold change of gene expression of tight junction markers as a proxy for barrier function in mice fed a HFD and intrarectally treated with PBS liposomes (blue) or clodronate liposomes (green). Top panel: small intestine, middle panel: distal colon, bottom panel: proximal colon.

References:

1. T. V. Rohm, R. Fuchs, R. L. Muller, L. Keller, Z. Baumann, A. J. T. Bosch, R. Schneider, D. Labes, I. Langer, J. B. Pitz, J. H. Niess, T. Delko, P. Hruz, C. Cavelti-Weder, Obesity in Humans Is Characterized by Gut Inflammation as Shown by Pro-Inflammatory Intestinal Macrophage Accumulation. *Front Immunol* **12**, 668654 (2021).
2. S. Traub, D. T. Meier, F. Schulze, E. Dror, T. M. Nordmann, N. Goetz, N. Koch, E. Dalmas, M. Stawiski, V. Makshana, F. Thorel, P. L. Herrera, M. Boni-Schnetzler, M. Y. Donath, Pancreatic alpha Cell-Derived Glucagon-Related Peptides Are Required for beta Cell Adaptation and Glucose Homeostasis. *Cell reports* **18**, 3192-3203 (2017).
3. K. Timper, E. Dalmas, E. Dror, S. Rutti, C. Thienel, N. S. Sauter, K. Bouzakri, B. Bedat, F. Pattou, J. Kerr-Conte, M. Boni-Schnetzler, M. Y. Donath, Glucose-Dependent Insulinotropic Peptide Stimulates Glucagon-Like Peptide 1 Production by Pancreatic Islets via Interleukin 6, Produced by alpha Cells. *Gastroenterology* **151**, 165-179 (2016).
4. C. C. Bain, C. L. Scott, H. Uronen-Hansson, S. Gudjonsson, O. Jansson, O. Grip, M. Williams, B. Malissen, W. W. Agace, A. M. Mowat, Resident and pro-inflammatory macrophages in the colon represent alternative context-dependent fates of the same Ly6Chi monocyte precursors. *Mucosal Immunol* **6**, 498-510 (2013).

REVIEWERS' COMMENTS:

Reviewer #1 (Remarks to the Author):

Accept

Reviewer #2 (Remarks to the Author):

The authors have addressed all comments and questions thoughtfully and thoroughly. Recommend accepting manuscript.

Point-by-point response to the reviewers comments

REVIEWERS' COMMENTS:

Reviewer #1 (Remarks to the Author):

Accept

Reviewer #2 (Remarks to the Author):

The authors have addressed all comments and questions thoughtfully and thoroughly. Recommend accepting manuscript.

Response: We are pleased that our revised manuscript answered all questions and concerns.